# Byzantine-Robust and Hessian-Free Federated Bilevel Optimization

**Shruti Maralappanavar**                                    *mshruti32@gmail.com*
*Department of Electrical, Electronics and Communication Engineering*
*Indian Institute of Technology Dharwad*
*Dharwad, India*

**B. N. Bharath**                                            *bharathbn@iitdh.ac.in*
*Department of Electrical, Electronics and Communication Engineering*
*Indian Institute of Technology Dharwad*
*Dharwad, India*

**Reviewed on OpenReview:** *https://openreview.net/pdf?id=5trmyvtkeo*

## Abstract

In the last few years, Byzantine robust algorithms to solve a minimization problem in the Federated setup have received significant attention. Most of the existing works consider the problem of byzantine-robustness for single-level optimization or consider the federated bilevel optimization without Byzantine nodes. However, problem formulation such as federated bilevel optimization in the presence of byzantine nodes is unexplored. Recognizing the gap, for the first time, we propose a computationally efficient and robust algorithm for solving Federated Bilevel Optimization with Byzantine (FedBOB) nodes that: ① Work under the assumption that the data across nodes are heterogeneous (non-iid), ② Consider the lower-level objective is non-convex and satisfies the Polyak-Łojasiewicz (PL)-inequality, and ③ Is fully first-order and does not rely on second order information. We achieve this by reformulating the federated bilevel problem into a single penalty problem. We provide the theoretical performance of the proposed algorithm and experimentally corroborate our theoretical findings.

## 1 Introduction

Deep Learning thrives on large datasets that are often distributed across multiple owners (Verbraeken et al., 2020). The challenge of training the model on distributed data sets was addressed using a paradigm called Federated Learning (FL), which enables clients (agents) to train models locally on private data while a central server (aggregator) combines them into a unified model (McMahan et al., 2017; Smith et al., 2017). Although FL offers several benefits, it comes with certain risks. Since it is a collaborative mechanism, it opens the possibility of security threats (Karimireddy et al., 2020; Gorbunov et al., 2022). In particular, a few nodes in the FL setting can potentially be malicious, also known as Byzantine nodes, and send corrupt information which renders the updates at the central server useless. Recently, the question of byzantine robustness has received significant attention making it relatively well studied both in theory and practice for single-level minimization problems in the FL setting (Yin et al., 2018; Blanchard et al., 2017b; Chen et al., 2017; Karimireddy et al., 2021; 2020; Rammal et al., 2024). There are many real-world applications that cannot be modeled as single level minimization problems, e.g. robust learning Zhang et al. (2022); Khanduri et al. (2023), meta-learning Rajeswaran et al. (2019), hyperparameter optimization Franceschi et al. (2018), neural architecture search Xue et al. (2021), resource management Sun et al. (2021), and image denoising Crockett et al. (2022) and other problems encountered in machine learning and signal processing tasks Zhang et al. (2024), to name a few. Such problems follow a nested structure that goes beyond the scope of a standard single-level minimization structure, as in Karimireddy et al. (2021; 2020); Rammal et al.

(2024). Towards solving such nested problems, bilevel optimization in non-FL as well as the FL setting has received great attention in the recent past Ghadimi & Wang (2018); Hong et al. (2023); Khanduri et al. (2021); Tarzanagh et al. (2022); Huang et al. (2023); Yang et al. (2024b). But these works cannot handle the presence of Byzantine clients. Recently, the authors in Abbas et al. (2024) proposed a byzantine-resilient bilevel federated optimization algorithm. However, they ① use second order information to update its model parameters making it computationally expensive, and ② make restrictive assumption such as strongly-convex lower level function.

A typical approach to solving the bilevel problems is to compute the gradient of the upper-level function (called hypergradient). To find the hypergradient there is a need to compute the second-order information of the lower-level function that makes the algorithm computationally complex (Ghadimi & Wang, 2018; Chen et al., 2021). This issue was resolved by using the first order method involving penalty formulation (Liu et al., 2022a; Kwon et al., 2023b; Shen & Chen, 2023; Chen et al., 2023) of the bilevel problem; albeit in the non-FL setting. However, none of the existing work considers a computationally efficient robust algorithm to solve a bilevel optimization problem in the FL setting leading to the following question.

> Q: *Is it possible to design a Robust and Low Complexity algorithm for solving a bilevel optimization problem in the FL setting in the presence of Byzantine nodes?*

In this paper, we answer the above question by considering a federated version of the bilevel optimization problem in the presence of Byzantine nodes. We assume that there are $N$ nodes denoted by $\mathcal{N} = \mathcal{G} \bigcup \mathcal{B}$, where $\mathcal{G}$ is the set of good or legitimate nodes, and $\mathcal{B}$ is the set of Byzantine or bad nodes. Let $|\mathcal{G}| = G$ and $|\mathcal{B}| = B$. The problem of Federated Bilevel Optimization with Byzantine (`FedBOB Problem`) involves solving the following:

$$\min_{\mathbf{x} \in \mathbb{R}^{d_1}} f(\mathbf{x}) \coloneqq f(\mathbf{x}, \mathbf{y}^*(\mathbf{x})) \coloneqq \frac{1}{G} \sum_{k \in \mathcal{G}} f_k(\mathbf{x}, \mathbf{y}^*(\mathbf{x}))$$

subject to

$$\mathbf{y}^*(\mathbf{x}) \in \left\{ \arg \min_{\mathbf{y} \in \mathbb{R}^{d_2}} g(\mathbf{x}, \mathbf{y}) \coloneqq \frac{1}{G} \sum_{k \in \mathcal{G}} g_k(\mathbf{x}, \mathbf{y}) \right\}, \tag{1}$$

where $f_k : \mathbb{R}^{d_1} \times \mathbb{R}^{d_2} \to \mathbb{R}$ and $g_k : \mathbb{R}^{d_1} \times \mathbb{R}^{d_2} \to \mathbb{R}$, $k \in \mathcal{G}$ are upper and lower level objective functions, respectively. The standard approach to solving the above problem involves computing second-order information such as Hessian, which is computationally expensive. In this paper, we take the first order approach by minimizing a penalty function defined as $\min_{\mathbf{x},\mathbf{y}} \{h_\lambda(\mathbf{x}, \mathbf{y}) \coloneqq f(\mathbf{x}) + \lambda p(\mathbf{x}, \mathbf{y})\}$, where $\lambda > 0$ and $p(\mathbf{x}, \mathbf{y}) \coloneqq g(\mathbf{x}, \mathbf{y}) - \min_{y'} g\left(\mathbf{x}, \mathbf{y}'\right)$ (Shen & Chen, 2023; Liu et al., 2022a; Kwon et al., 2023b;a). Next, we present challenges that we need to address in comparison with the existing work (Karimireddy et al., 2020; Shen & Chen, 2023; Blanchard et al., 2017a; Pillutla et al., 2022a).

**Challenges:** The penalty based method in the non-FL setting is very well studied (Shen & Chen, 2023). This work relies on the equivalence between a local solution to the penalty formulation (i.e., the point at which the gradient is zero) and a solution to the original problem. Unfortunately, the FL setting with heterogeneous and Byzantine nodes forces the solution to have non-zero gradients (see lower bounds in (Karimireddy et al., 2020)), and hence the equivalence in (Shen & Chen, 2023) cannot be directly used. We take a different approach where we analyze (i) convergence of the gradient, as in (Shen & Chen, 2023) and (b) a new notion called constraint violation that measures the average violation of the constraint in the lower-level problem, which is new. Keeping track of the average constraint violation is particularly important in settings such as adversarial learning (Khanduri et al., 2023). The average constraint violation gives information on how frequently the constraint is violated rather than capturing it implicitly in the convergence rate, as in (Shen & Chen, 2023). In addition, minimizing the penalty function involves computing $\min_{y'} g\left(\mathbf{x}, \mathbf{y}'\right)$ in a FL fashion in the presence of Byzantine. To make matters more challenging, the resulting lower-level optimum drifts away from $\mathbf{y}^*(\mathbf{x})$ due to heterogeneity in the data and is further alleviated due to the presence of Byzantine nodes, which makes the analysis difficult.

**Contribution:** In this paper, we address the above challenges and make the following contribution.
① We consider the FedBOB problem in equation 1, and propose a robust and fully first-order federated al-

gorithm by reformulating the federated bilevel problem into a single-level penalty-based optimization problem that makes it computationally efficient. Unlike the existing work on federated bilevel optimization (Tarzanagh et al., 2022; Huang et al., 2023; Yang et al., 2024b) where the lower level is strongly convex, we consider a class of non-convex functions satisfying the PL inequality, thus widening the scope of applicability.

② The proposed algorithm (Rob-FedBOB) is shown to converge at a rate of $\mathcal{O}\left(\frac{\lambda^2}{R} + \alpha(\zeta_f^2 + \lambda^2\zeta_g^2)\right)$, where $R$ is the number of communication rounds, $\zeta_f$ and $\zeta_g$ are the inter-client heterogeneity terms corresponding to the upper and lower level objective functions, respectively, and $\alpha \in [0, 0.5]$ is the fraction of the Byzantine nodes. Additionally, the algorithm results in an average constraint violation that scales as $\mathcal{O}\left(\frac{(1+c\alpha\zeta_f^2)}{\lambda^2} + c\alpha\zeta_g^2\right)$. Our results demonstrate the trade-off between the convergence and the constraint violation that is captured through $\lambda$. Higher $\lambda$ results in better constraint violation properties, but at the expense of lower convergence rate and vice versa. Furthermore, our bounds reveal that Byzantine nodes and heterogeneity act as a bottleneck in achieving good performance, as expected. We study the trade-off between convergence and the constraint violation, and provide insight on the choice of $\lambda$.

③ Finally, we present experimental results for data hyperclearning application for various attacks and corroborate our theoretical findings. In particular, we consider the following attacks (i) Bit Flipping (BF), (ii) Label Flipping (LF), (iii) Inner Product Manipulation (IPM), and (iv) A Little is Enough (ALIE), and show that both gradient and constraint violation go down and converge to a constant with increasing number of communication rounds. The constants to which the gradient and the average violation converge are governed by the number of Byzantine nodes, and the inter-client heterogeneity of the good nodes.

## 1.1 Related Work

**Robust Federated Learning:** Over the recent years there has been a significant amount of work on byzantine robustness in case of single level optimization problems (Karimireddy et al., 2020; Gorbunov et al., 2022). Byzantine robustness is very well studied when the nodes have iid data distributions. One approach to obtaining robustness is to use robust aggregation strategies such as Coordinate wise median (Chen et al., 2017), KRUM (Blanchard et al., 2017a), geometric median (Pillutla et al., 2022a), use variance reduction techniques (Wu et al., 2020), filter byzantine nodes. Recently, there have been many works which consider byzantine robustness for non-iid data distributions such as: outlier based-robust clustering (Sattler et al., 2020), spectral methods for robust optimization (Data & Diggavi, 2021). Other interesting approaches include use of (a) a formal definition of robust aggregation with client momentum (Karimireddy et al., 2020), (b) random checks of computations (Rammal et al., 2024), (c) normalized gradient (Zuo et al., 2024), and (d) normalized momentum (Yang et al., 2024a). These works are limited to solving single level federated learning problem in the presence of Byzantine nodes.

**Federated Bilevel Optimization (FBO):** Recently some works have considered Federated Bilevel optimization as many machine learning applications can be formulated as a nested bilevel problem. The authors in (Tarzanagh et al., 2022) proposed a federated alternating stochastic gradient method that requires a federated hypergradient computation. They use variance reduction to handle lower-level heterogeneity. The results on the complexity of the sample were improved using momentum-based federated bilevel algorithms with a reduction in variance in (Li et al., 2022). Later, the work in (Huang et al., 2023) achieved linear speedup in the presence of non-iid data by using a novel client sampling scheme. Along similar lines, the authors in (Yang et al., 2024b) propose a communication efficient federated bilevel optimization algorithm. However, most of these works require the second-order information to perform the gradient update. Further, these works limit the lower-level objective function to strongly convex. Thus, the challenging problem of finding a computationally efficient algorithm that is robust to byzantine attacks in a federated bilevel setting is still an open problem. In this work, we have closed this gap. Concurrent to our work, the authors in (Abbas et al., 2024) proposed a byzantine-resilient bilevel federated optimization algorithm. They use the second order information such as the Hessians/Jacobians making it computationally expensive. In addition, we have included more experimental results compared to (Abbas et al., 2024).

## 2 Problem Statement

We consider a federated version of the bilevel optimization problem in the presence of Byzantine nodes as in equation 1. The problem in equation 1 cannot be solved directly in the presence of Byzantine nodes. This is due to the fact that the Byzanine nodes can share corrupt information with the central server, which can make the updates potentially useless. The problem of Byzantine robust distributed optimization in various settings have been studied earlier (Blanchard et al., 2017a; Pillutla et al., 2022a; Wu et al., 2020; He et al., 2020). Clearly, as explained in the previous section, the problem of federated bilevel optimization with Byzantine is not at all addressed in the literature as it posses several challenges: ① The typical solution for the bilevel problem without Byzantine involves using second order methods thus making it computationally expensive (Ghadimi & Wang, 2018; Chen et al., 2021). ② Most of these works assume that the lower level function $g_k(\mathbf{x}, \mathbf{y})$ for all $k \in \mathcal{G}$ is strongly convex making it more restrictive. ③ The lower level optimum drifts away from $\mathbf{y}^*(\mathbf{x})$ due to heterogeneity in the data, and is further aggravated due to the presence of Byzantine nodes. We handle the above challenges by ① proposing a fully first order method of solving `FedBOB Problem`, and ② using robust aggregation strategies that is resilient to the Byzantine attacks. Our method is motivated by reformulating the `FedBOB Problem` into an approximate equivalent form as follows:

$$\texttt{Approx-FedBOB Problem:} \min_{\mathbf{x}} \ f(\mathbf{x}, \mathbf{y})$$
$$\text{such that } p(\mathbf{x}, \mathbf{y}) \coloneqq g(\mathbf{x}, \mathbf{y}) - v(\mathbf{x}) \leq \epsilon, \tag{2}$$

where $v(\mathbf{x}) \coloneqq \min_y g(\mathbf{x}, \mathbf{y})$ and $\epsilon > 0$. Letting $\epsilon = 0$ in the above problem makes it equivalent to that of the `FedBOB Problem`. Writing the Lagrangian function of the above problem results in

$$h_\lambda(\mathbf{x}, \mathbf{y}) \coloneqq \frac{1}{G} \sum_{k=1}^{G} h_{\lambda,k}(\mathbf{x}, \mathbf{y}) \coloneqq f(\mathbf{x}, \mathbf{y}) + \lambda p(\mathbf{x}, \mathbf{y}) \tag{3}$$

for all $\mathbf{x} \in \mathbb{R}^{d_1}$, $\mathbf{y} \in \mathbb{R}^{d_2}$. In the above, $\lambda > 0$ and $p(\mathbf{x}, \mathbf{y})$ is the penalty term. Now, the dual problem is to solve the following in the presence of Byzantine nodes

$$\texttt{FedBOB Penalty Problem:} \min_{\mathbf{x}, \mathbf{y}} \ h_\lambda(\mathbf{x}, \mathbf{y}). \tag{4}$$

It turns out that solving the above problem is approximately equivalent to solving the original problem (see (Shen & Chen, 2023) for more details). Note that this boils down to a single level federated learning problem in the presence of Byzantine. Although single level federated optimization in the presence of Byzantine (see (Karimireddy et al., 2020; Gorbunov et al., 2022; Pillutla et al., 2022a; Wu et al., 2020)) has been studied in the literature, our framework is completely different in the following sense ① the impact of the Byzantine nodes and $\lambda$ on the performance need to be studied, and does not follow directly from the existing literature. ② The solution obtained by our proposed algorithm should not only result in a good stationary point of equation 4 but also should exhibit good constraint violation properties—this additional requirement is not explicit in the single level formulation. Proposing a low complexity robust algorithm and analyzing its performance is completely new.

### 2.1 Preliminaries

In this subsection, we discuss the assumptions and definitions required in the analysis of the proposed Robust Federated Bilevel Optimization with Byzantine (Rob-FedBOB) algorithm.

**Assumption 1.** *(Smoothness): The upper-level objective function $f_k(\mathbf{x}, \mathbf{y})$ is assumed to be $L_{f,k}$ smooth for all good nodes $k \in \mathcal{G}$, i.e.,*

$$\|\nabla f_k(\mathbf{x}, \mathbf{y}) - \nabla f_k(\mathbf{x}, \mathbf{y}')\|_2 \leq L_{f,k} \|\mathbf{y} - \mathbf{y}'\|_2$$

*for all $\mathbf{x}, \mathbf{x}' \in \mathbb{R}^{d_1}$, $\mathbf{y}, \mathbf{y}' \in \mathbb{R}^{d_2}$. Further, we also assume that $f_k(\mathbf{x}, \mathbf{y})$ is a $l_{f,k}-Lipschitz$, i.e.,*

$$| f_k(\mathbf{x}, \mathbf{y}) - f_k(\mathbf{x}, \mathbf{y}') | \leq l_{f,k} \|\mathbf{y} - \mathbf{y}'\|_2$$

*for all $\mathbf{x}, \mathbf{x}' \in \mathbb{R}^{d_1}$, $\mathbf{y}, \mathbf{y}' \in \mathbb{R}^{d_2}$ and $k \in \mathcal{G}$. In addition, $g_k(\mathbf{x}, *)$ is a $L_{g,k}$-smooth and $l_{g,k}-lipschitz$ function for all $k \in \mathcal{G}$.*

**Assumption 2.** *(PL inequality): The lower-level objective function $g_k(\mathbf{x}, \mathbf{y})$ for all $k \in \mathcal{G}$ satisfies the PL inequality, i.e., $\|\nabla g_k(\mathbf{x}, \mathbf{y})\|^2 \geq \mu_g(g_k(\mathbf{x}, \mathbf{y}) - v_k(\mathbf{x}))$ for some $\mu_g > 0$ and for all $\mathbf{x} \in \mathbb{R}^{d_1}, \mathbf{y}_1, \mathbf{y}_2 \in \mathbb{R}^{d_2}$. Here $y_k^*(\mathbf{x}) \in \min_{\mathbf{y}} g_k(\mathbf{x}, \mathbf{y})$. In addition, we consider the average lower level objective $g(\mathbf{x}, \mathbf{y})$ satisfies the PL inequality, i.e., $\|\nabla g(\mathbf{x}, \mathbf{y})\|^2 \geq \mu_g(g(\mathbf{x}, \mathbf{y}) - v(\mathbf{x}))$ for some $\mu_g > 0$ for all $\mathbf{x} \in \mathbb{R}^{d_1}, \mathbf{y} \in \mathbb{R}^{d_2}$.*

**Assumption 3.** *(Inter-client heterogeneity): The upper-level objective function $f_k(\mathbf{x}, \mathbf{y})$ for all $k \in \mathcal{G}$ is said to satisfy inter-client heterogeneity, i.e., $\mathbb{E}_{k \in \mathcal{G}} \|\nabla f_k(\mathbf{x}, \mathbf{y}) - \nabla f(\mathbf{x}, \mathbf{y})\|^2 \leq \zeta_f^2$ for some $\zeta_f > 0$ and for all $\mathbf{x} \in \mathbb{R}^{d_1}, \mathbf{y} \in \mathbb{R}^{d_2}$. In addition, $g_k(\mathbf{x}, \mathbf{y})$ also satisfies inter-client heterogeneity for some $\zeta_g > 0$, i.e., $\mathbb{E}_{k \in \mathcal{G}} \|\nabla g_k(\mathbf{x}, \mathbf{y}) - \nabla g(\mathbf{x}, \mathbf{y})\|^2 \leq \zeta_g^2$.*

Most of the assumptions above such as smoothness, inter-client heterogeneity are standard (Shen & Chen, 2023; Karimireddy et al., 2020; Rammal et al., 2024). The PL inequality is satisfied by most of the over-parameterized neural networks (Liu et al., 2022b; Shen & Chen, 2023). The inter-client heterogeneity (Karimireddy et al., 2020; Gorbunov et al., 2022; Yu et al., 2018) restricts the data heterogeneity of good nodes. In fact, the lower bound in (Karimireddy et al., 2020) shows that the heterogeneity condition is inevitable. Note that $\zeta_f = 0$ and $\zeta_g = 0$ case imply that the data are homogeneous across all nodes.

## 3 Algorithm Design

It is well known that the aggregator plays an important role in the performance of any algorithm with Byzantines. Hence, we first define the class of aggregators, which is adapted from (Karimireddy et al., 2020).

**Definition 1.** *$((\alpha, c)$-Robust Aggregator): Let us assume that we are given inputs $\{\mathbf{x}_1, \mathbf{x}_2, \ldots, \mathbf{x}_N\}$ such that there exists a subset $\mathcal{G} \subseteq [N]$ of size $\mid \mathcal{G} \mid = G \geq (1 - \alpha)N$ for $\alpha \leq 0.5$ such that $\mathbb{E} \|\mathbf{x}_i - \mathbf{x}_j\|^2 \leq \delta^2$ for all $i, j \in \mathcal{G}$ and some $\delta \geq 0$. Then we say that $\hat{\mathbf{x}}$ satisfies $(\alpha, c)$-Robust Aggregator (RAgg) if*

$$\mathbb{E} \|\hat{\mathbf{x}} - \bar{\mathbf{x}}\|^2 \leq c\alpha\delta^2, \tag{5}$$

*where $\bar{\mathbf{x}} = \frac{1}{G} \sum_{k=1}^{G} \mathbf{x}_k$. Here the expectation is taken with respect to possible randomness of good nodes $\{\mathbf{x}_1, \mathbf{x}_2, \ldots, \mathbf{x}_G\}$.*

Note that $(\alpha, c)$-Robust Aggregator property is satisfied by many aggregators such as KRUM, Coordinate-wise median (CM), Geometric median (RFA). We provide more details in Appendix H.

### 3.1 Rob-FedBOB Algorithm

Given that the penalty reformulation is a single level problem, an obvious approach is to use the following gradient updates: $\mathbf{x}_{r+1} = \mathbf{x} - \beta \nabla_{\mathbf{x}} h_\lambda(\mathbf{x}_r, \mathbf{y}_r)$ and $\mathbf{y}_{r+1} = \mathbf{y} - \beta \nabla_{\mathbf{y}} h_\lambda(\mathbf{x}_r, \mathbf{y}_r)$. However, this computation involves finding $\nabla p(\mathbf{x}_r, \mathbf{y}_r)$ which requires $\nabla v(\mathbf{x}_r)$. In general, $v(\mathbf{x})$ need not to smooth always. Further, $\nabla_{\mathbf{x}} v(\mathbf{x}) \neq \nabla_{\mathbf{x}} g(\mathbf{x}, \mathbf{y}^*)$ at $\mathbf{y}^*$. From assumptions 2 and 1, using Lemma A.5 of Nouiehed et al. (2019), we can see that $\nabla v(\mathbf{x}) = \nabla_{\mathbf{x}} g(\mathbf{x}, \mathbf{y}^*(\mathbf{x}))$. In addition to this, the gradient update requires $\mathbf{y}^*(\mathbf{x})$, which is a solution to the lower level problem with respect to $\mathbf{y}$, and is unknown. More specifically, computing $\mathbf{y}^*(\mathbf{x})$ requires access to $g(\mathbf{x}, \mathbf{y})$ (see equation 1), which is not available at each node. One way to handle this is that each node $k$ runs $T$ number of GD steps on the lower level function $g_k(\mathbf{x}_r, \mathbf{y})$ for a given $\mathbf{x}_r$ as follows (see line 9 of Algorithm 1)

$$\mathbf{y}_{r,t+1} = \mathbf{y}_{r,t} - \gamma \nabla_{\mathbf{y}} g_{\mathbf{ag}}(\mathbf{x}_r, \mathbf{y}_{r,t}),$$

where $\mathbf{y}_{r,0} = \mathbf{y}_{r-1}$ and $t = 0, 1, \ldots, T - 1$. Here,

$$\nabla_{\mathbf{y}} g_{\mathbf{ag}}(\mathbf{x}_r, \mathbf{y}_r) = \text{RAgg}(\nabla_{\mathbf{y}} g_k(\mathbf{x}_r, \mathbf{y}_r), k \in [N]), \tag{6}$$

where RAgg uses bucketing followed by geometric median aggregation (see (Karimireddy et al., 2020) for more details). Now, we can use $\mathbf{y}_{k,T}$ as a proxy for $\mathbf{y}^*(\mathbf{x}_r)$ to get the following updates (see line 13 of

Algorithm 1):

$$\mathbf{y}_{r+1} = \mathbf{y}_r - \eta \nabla_{\mathbf{y}} h_{\lambda,\mathbf{ag}}(\mathbf{x}_r, \mathbf{y}_r), \text{ and} \tag{7}$$

$$\mathbf{x}_{r+1} = \mathbf{x}_r - \beta \nabla_{\mathbf{x}} \hat{h}_{\lambda,\mathbf{ag}}(\mathbf{x}_r, \mathbf{y}_r), \tag{8}$$

where the gradients are defined as

$$\nabla_{\mathbf{y}} h_{\lambda,\mathbf{ag}}(\mathbf{x}_r, \mathbf{y}_r) = \mathtt{RAgg}\left(\nabla_{\mathbf{y}} h_{\lambda,k}(\mathbf{x}_r, \mathbf{y}_r), k \in [N]\right),$$

$$\nabla_{\mathbf{x}} \hat{h}_{\lambda,\mathbf{ag}}(\mathbf{x}_r, \mathbf{y}_r) = \mathtt{RAgg}\left(\nabla_{\mathbf{x}} \hat{h}_{\lambda,k}(\mathbf{x}_r, \mathbf{y}_r), k \in [N]\right).$$

In the above,

$$\nabla_{\mathbf{x}} \hat{h}_{\lambda,k}(\mathbf{x}_r, \mathbf{y}_r) := \nabla_{\mathbf{x}} f_k(\mathbf{x}_r, \mathbf{y}_r) + \lambda\left(\nabla_{\mathbf{x}} g_k(\mathbf{x}_r, \mathbf{y}_r) - \nabla_{\mathbf{x}} g_k(\mathbf{x}_r, \mathbf{y}_{r,T})\right) \tag{9}$$

and $\nabla_{\mathbf{y}} h_{\lambda,k}(\mathbf{x}_r, \mathbf{y}_r) := \nabla_{\mathbf{y}} f_k(\mathbf{x}_r, \mathbf{y}_r) + \lambda \nabla_{\mathbf{y}} g_k(\mathbf{x}_r, \mathbf{y}_r)$ for all $k \in \mathcal{G}$. In order to access whether the output of the algorithm satisfies the constraint or not, we propose the following notion of violation:

$$\mathtt{Viol}_R := \sum_{r=0}^{R-1} p(\mathbf{x}_r, \mathbf{y}_r), \tag{10}$$

where $\mathbf{x}_r$ and $\mathbf{y}_r$ are the output of the algorithm in round $r$. Note that when $\mathtt{Viol}_R/R$ converges to a small constant, it means that $p(\mathbf{x}_r, \mathbf{y}_r)$ for sufficiently large $r$ is close to the constant, and hence captures the violation performance of the algorithm.

**Note:** The robust strategy involving binning followed by geometric median is known to satisfy the $(\alpha, c)$ robustness (see (Karimireddy et al., 2020)) property. However, in our case, the gradient of the penalty function is aggregated at the central server. This function depends on $\lambda$ and the proxy for $\mathbf{y}^*(\mathbf{x}_r)$, which is obtained by using lines 5 to 11 in Algorithm 1. This makes it necessary for us to prove that the $(\alpha, c)$ robustness is still achieved for the penalty function, as shown in the following lemmas.

---

**Lemma 1.** *For the robust aggregator* `RAgg`*, we have for some* $c > 0$

$$\left\| \nabla_{\mathbf{y}} h_{\lambda,ag}(\mathbf{x}_r, \mathbf{y}_r) - \nabla_{\mathbf{y}} h_{\lambda}(\mathbf{x}_r, \mathbf{y}_r) \right\|^2 \leq c\alpha\rho^2$$

*where* $\rho^2 := 8\zeta_f^2 + 8\lambda^2 \zeta_g^2$.

---

*Proof:* See Sec. B in Appendix. $\qquad\square$

The above lemma shows that the robustness depends on $\lambda$ and the heterogeneity terms. In the following, we prove the robustness of $\nabla_{\mathbf{x}} \bar{\hat{h}}_{\lambda}(\mathbf{x}_r, \mathbf{y}_r) := \frac{1}{G} \sum_{k=1}^{G} \nabla_{\mathbf{x}} \hat{h}_{\lambda,k}(\mathbf{x}_r, \mathbf{y}_r)$ with $\delta^2 := \frac{8\lambda^2 L_{g,\max}^2 l_{g,\max}^2}{\mu_g^2}\left(1 - \frac{\gamma\mu_g}{2}\right)^T + \frac{8\lambda^2 L_{g,\max}^2 c\alpha \zeta_g^2}{\mu_g} + 6\zeta_f^2 + 12\lambda^2\zeta_g^2$, where $L_{g,max} := \max_{k\in\mathcal{G}} L_{g,k}$ and $l_{g,max} := \max_{k\in\mathcal{G}} l_{g,k}$.

---

**Lemma 2.** *The robust aggregator* `ARgg` *satisfies*

$$\left\| \nabla_{\mathbf{x}} \hat{h}_{\lambda,ag}(\mathbf{x}_r, \mathbf{y}_r) - \nabla_{\mathbf{x}} \bar{\hat{h}}_{\lambda}(\mathbf{x}_r, \mathbf{y}_r) \right\|^2 \leq c\alpha\delta^2.$$

---

*Proof:* See Sec. C in Appendix. $\qquad\square$

The above lemma shows that the robustness depends on how close the proxy $\mathbf{y}_{r,T}$ is to the actual optimal $\mathbf{y}^*(\mathbf{x}_r)$ captured through the first term in $\delta^2$. It also reveals that this can be reduced by increasing $T$.

---

**Algorithm 1** Rob-FedBOB Algorithm

---

1: **Initialize** $\mathbf{x}_0 \in \mathbb{R}^{d_1}$, $\mathbf{y}_0 \in \mathbb{R}^{d_2}$
2: **for** $r = 0, 1, 2, \ldots, R - 1$ **do**
3:     **Send** $\mathbf{x}_r, \mathbf{y}_r$ to each node
4:     **Set** $\mathbf{z}_{r,0} = \mathbf{y}_r$
5:     **for** $t = 0, 1, \ldots, T - 1$ **do**
6:         **for** $k \in \mathcal{N}$ **in parallel do**
7:             **Send** $\nabla_{\mathbf{y}} g_k (\mathbf{x}_r, \mathbf{y}_{r,t})$, $k \in \mathcal{N}$
8:         **end for**
9:         $\mathbf{y}_{r,t+1} = \mathbf{y}_{r,t} - \gamma \nabla_{\mathbf{y}} g_{\mathbf{ag}} (\mathbf{x}_r, \mathbf{y}_{r,t})$
10:        **Send** $\mathbf{y}_{r,t+1}$ to all nodes
11:    **end for**
12:    **Set** $\mathbf{y}_{r,T} = \mathbf{y}_{r,T}$
13:    **Server updates** $\mathbf{x}_{r+1}$ and $\mathbf{y}_{r+1}$ using equation 8 and equation 7, respectively.
14: **end for**
     **Output:** $\mathbf{x}_R$ and $\mathbf{y}_R$

---

## 3.2 Convergence Analysis

In the following, we provide the first main result for the Rob-FedBOB algorithm.

---

**Theorem 1.** *Suppose assumptions 1-3 hold, then for the aggregator* `RAgg`*, Algorithm 1 achieves the following bound*

$$\frac{1}{R} \sum_{r=0}^{R-1} \|\nabla h_\lambda (\mathbf{x}_r, \mathbf{y}_r)\|^2 \leq \mathcal{O} \left( \frac{\lambda^2}{R} + \alpha(\zeta_f^2 + \lambda^2 \zeta_g^2) \right)$$

*for constant learning rates* $\eta \leq \frac{1}{L_h}$, $\beta \leq \frac{1}{L_h}$*, and* $T \geq \frac{2}{\gamma \mu_g} \log \left( \frac{8 \lambda^2 R L_{g,\max}^2 l_{g,\max}^2}{\mu_g^2} \right)$*.*

---

*Proof:* See Sec. F in Appendix. □

The above result reveals the effect of $\lambda$, the fraction of the Byzantine nodes $\alpha$, and the heterogeneity terms $\zeta_f$ and $\zeta_g$ on the convergence rate. Note that when $\alpha = 0$, i.e., when there are no byzantine nodes, we get a convergence rate of $\mathcal{O} \left( \frac{\lambda^2}{R} \right)$. Thus, a smaller gradient can be achieved by choosing larger $R$. On the other hand, in the presence of Byzantine, i.e., $\alpha \neq 0$, we have

$$\frac{1}{R} \sum_{r=0}^{R-1} \|\nabla h_\lambda (\mathbf{x}_r, \mathbf{y}_r)\|^2 = \mathcal{O}(\alpha(\zeta_f^2 + \zeta_g^2)). \tag{11}$$

This cannot be made very small unless the heterogeneity terms $\zeta_f^2$ and $\zeta_g^2$ are zeros. A similar observation was made in Karimireddy et al. (2020) in the context of single level setting. They also showed a lower bound demonstrating that the result they obtain cannot be improved further. Since single level is a special case of the bilevel problem that we are considering, we believe that the bound in Theorem 1 cannot be improved. Next, we present a bound on the average violation that relates to the gradient of the penalty function.

---

**Lemma 3.** *Suppose Assumptions 1 and 2 hold, and* $\|\nabla h_\lambda (\mathbf{x}, \mathbf{y})\| \leq \psi$ *for some* $\psi > 0$*, then the average violation in equation 10 is bounded as follows*

$$\frac{Viol_R}{R} \leq \frac{2 \left( l_f^2 + \psi^2 \right)}{\mu_g \lambda^2}. \tag{12}$$

---

*Proof:* See Sec. D in Appendix.

Note that the above shows that by choosing a large $\lambda$ or if $\psi$ is small results in a good constraint qualification properties. In order to prove a violation bound on the proposed algorithm, we use a bound on $\|\nabla h_\lambda (\mathbf{x}_r, \mathbf{y}_r)\|^2$ for each $r$ obtained in the proof of Theorem 1.

---

**Theorem 2.** *Algorithm 1 satisfies the following average constraint violation provide the conditions in Theorem 1 are satisfied*

$$\frac{\mathit{Viol}_R}{R} \leq \mathcal{O}\left(\frac{(1 + c\alpha\zeta_f^2)}{\lambda^2} + c\alpha\zeta_g^2\right).$$

---

*Proof:* See Sec. G in Appendix.

We observe from the above theorem that larger values of $\lambda$ result in a better violation qualification properties, as expected. In particular, as $\lambda \to \infty$, average violation converges to $c\alpha\zeta_g^2$ revealing the bottleneck due to the Byzantine nodes and the heterogeneity in the lower level function. Since the violation is with respect to the lower level function, it is expected that the heterogeneity in the upper level function impacts mildly on the average violation. However, larger $\lambda$ results in a bad gradient bound leading to a trade-off between convergence (gradient bound) and violation.

**Convergence versus Violation trade-off:** To better understand the trade-off, consider $\alpha = 0$ condition, i.e., no Byzantine nodes. In this case, choosing $\lambda = \mathcal{O}\left(\frac{1}{\sqrt{\epsilon}}\right)$ results in an average violation of $\mathcal{O}(\epsilon)$ and the average gradient converges at the rate of $\mathcal{O}\left(\frac{1}{R}\right)$ recovering the results of (Huang et al., 2023; Tarzanagh et al., 2022) when the variance is zero. The presence of the Byzantine nodes (i.e., $\alpha \neq 0$) changes the scenario completely. Choosing $\lambda$ as above, and $R \to \infty$ results in an average violation of $\mathcal{O}\left(c\alpha\zeta_g^2\right)$ and the convergence rate of the average gradient decreases to

$$\lim_{R\to\infty} \frac{1}{R} \sum_{r=0}^{R-1} \|\nabla h_\lambda (\mathbf{x}_r, \mathbf{y}_r)\|^2 \leq \mathcal{O}\left(\alpha\left(\zeta_f^2 + \frac{\zeta_g^2}{\epsilon}\right)\right). \tag{13}$$

This also reveals the impact of heterogeneity in the presence of Byzantine nodes.

**Remark:** Note that the $((\alpha, c)$-Robust Aggregator, initially proposed by Karimireddy et al. (2021; 2020) plays an important role in our algorithm. Most of the existing works that use this robust aggregator (Karimireddy et al., 2020; Rammal et al., 2024; Yang et al., 2024a) with single-level optimization problem. However, we consider a bilevel problem, which requires us to carefully combine the upper and lower level objective functions. Moreover, we need to obtain the optimal solution $\mathbf{y}^*(\mathbf{x})$ to the lower level problem, which adds complexity to our analysis. Also, our problem formulation requires proving convergence of the gradient and a bound on the average constraint violation, which is completely new. In addition, $\lambda$ plays an important role in studying the trade-off between convergence of the gradient and the constraint violation guarantee; this makes the problem more challenging compared to existing work (Karimireddy et al., 2020; Shen & Chen, 2023).

**Complexity:** The total number of communication rounds for bilevel federated learning problems (with or without Byzantine) is $\mathcal{O}(R)$ (Huang et al., 2023). Our communication complexity is $\mathcal{O}(R \log R)$, and order-wise matches with other existing algorithms except for a $\log R$ factor; this is done to ensure a good proxy for $\mathbf{y}^*(\mathbf{x}_r)$ (see lines 5 to 11 of Algorithm 1). In contrast to the existing algorithms in the federated bilevel learning problem (Tarzanagh et al., 2022; Huang et al., 2023), we require only first-order information, and as a consequence, the run time at each node is very low as demonstrated in our experimental results.

### 3.3 Proof Sketch of Theorem 1

In order to prove our main results, we first need the smoothness of the penalty problem.

> **Lemma 4.** *Suppose assumption 1 hold, then the function $h(\mathbf{x}, \mathbf{y})$ is $L_h$ smooth i.e.,*
>
> $$\|\nabla h_\lambda(\mathbf{x}, \mathbf{y}) - \nabla h_\lambda(\mathbf{x}, \mathbf{y}')\| \leq L_h \|\mathbf{y} - \mathbf{y}'\|,$$
>
> *where $L_h := L_f + \lambda L_g$.*

*Proof:* See Sec. E in Appendix. □

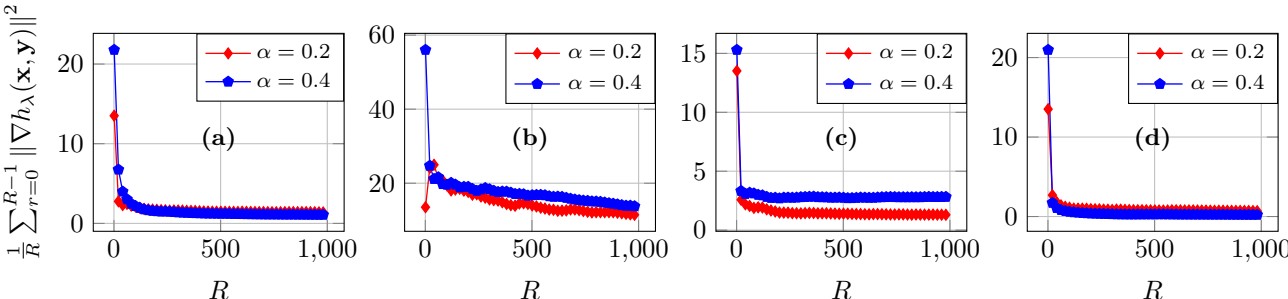

Figure 1: Effect of $\alpha$ on the convergence of Rob-FedBOB under BF (see (a)), LF (see (b)), IPM (see (c)) and ALIE (see (d)) attacks on the data hypercleaning application.

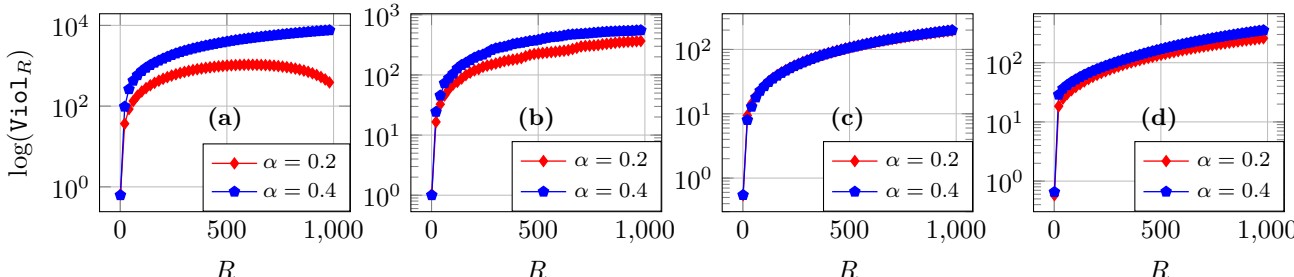

Figure 2: Effect of $\alpha$ on the violation under BF (see (a)), LF (see (b)), IPM (see (c)) and ALIE (see (d)) attacks on the data hypercleaning application.

Note that unlike vanilla FL problems, in the bilevel problems, we need extra rounds of communication (see lines 5 to 11 of the algorithm) to obtain a good proxy for the lower level problem, which is stated in the next lemma.

> **Lemma 5.** *Under assumptions 1-2 and choosing $\eta \leq \frac{1}{\mu_g}$, the approximation error in Algorithm 1 is bounded as*
> $$d^2_{S(\mathbf{x}_r)}(\mathbf{y}_{r,T}) \leq \frac{2l^2_{g,\max}}{\mu_g^2}\bar{\gamma}^T + \frac{2c\alpha\zeta_g^2}{\mu_g},$$
>
> *where $\bar{\gamma} := \left(1 - \frac{\gamma\mu_g}{2}\right)$.*

*Proof:* See Sec. E in Appendix. □

Note that by choosing $T$ as in Theorem 1 ensures that $d^2_{S(\mathbf{x}_r)}(\mathbf{y}_{r,T}) \leq \mathcal{O}\left(\frac{1}{R} + \frac{2c\alpha\zeta_g^2}{\mu_g}\right)$ indicating that the presence of Byzantines effect the proxy for the lower level problem as well. Since the gradients $\nabla_\mathbf{y} h_{\lambda,k}(\mathbf{x}_r, \mathbf{y}_r)$ and $\nabla_\mathbf{x} \hat{h}_{\lambda,k}(\mathbf{x}_r, \mathbf{y}_r)$ are sent to the central server, and is aggregated using bucketing followed by geometric median, we first need to relate these gradients to the true average gradients of the good nodes. This is done using Lemmas 1 and 2. Once these results are established, we prove a bound on $\|\nabla h_\lambda(\mathbf{x}, \mathbf{y})\|^2$, which is

summed over $r = 0, 1, \ldots, R-1$ to get a bound on the total gradient as in Theorem 1. We establish a result that connects violation to the bound on $\|\nabla h_\lambda(\mathbf{x}, \mathbf{y})\|^2$ as in Lemma 3. Using this Lemma, we establish the rates for the average violation as in Lemma 2.

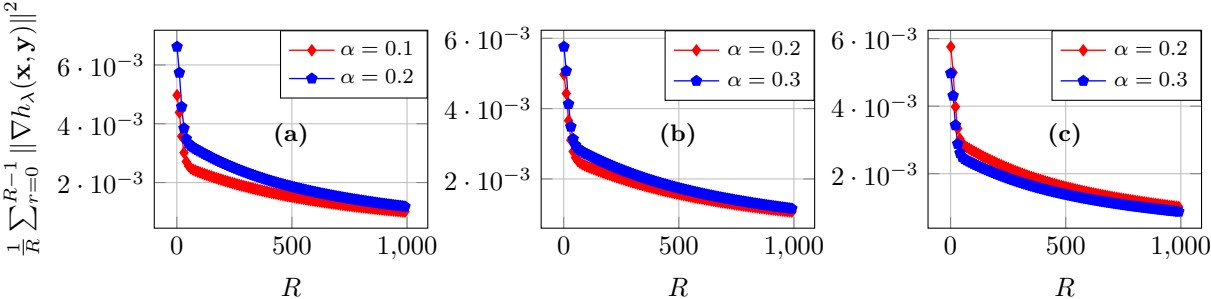

Figure 3: Effect of $\alpha$ on the convergence of Rob-FedBOB under BF (see (a)), IPM (see (b)) and ALIE (see (c)) attacks on the learnable regularization application.

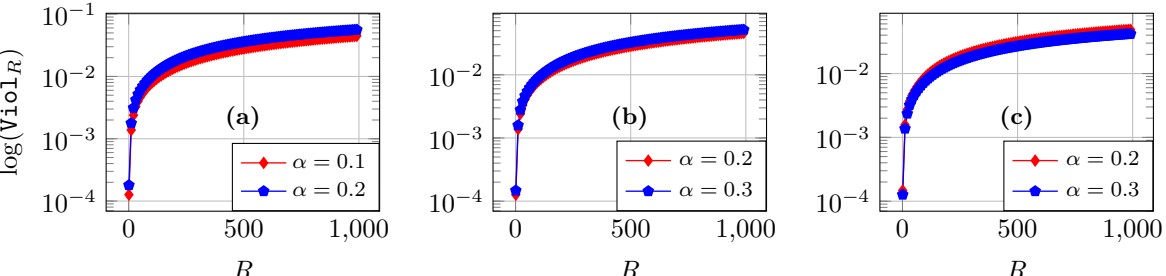

Figure 4: Effect of $\alpha$ on the violation under BF (see (a)), LF (see (b)), IPM (see (c)) and ALIE (see (d)) attacks on the learnable regularization application.

## 4 Experimental Results

In this section, we experimentally validate our theoretical findings of Rob-FedBOB. We evaluate the performance of Rob-FedBOB on two ML applications.

**Data Hyper-cleaning:** We consider the Data Hyper-cleaning task to experimentally validate our theory (Shaban et al., 2019; Kwon et al., 2023b) for the following attacks (i) BF, (ii) LF, (iii) IPM, and (iv) ALIE. The goal of Data Hyper-cleaning is to solve the following bilevel problem in a federated manner:

$$\min_{\mathbf{x}} \frac{1}{Gm} \sum_{k=1}^{G} \sum_{i=1}^{m} l_{k,i}\left(\mathbf{y}^*(\mathbf{x}); \mathcal{D}_k^{\text{val}}\right)$$

subject to

$$\mathbf{y}^*(\mathbf{x}) \in \arg\min_{\mathbf{y}} \frac{1}{Gn} \sum_{k=1}^{G} \sum_{i=1}^{n} \sigma(\mathbf{x}_{k,i}) l_{k,i}\left(\mathbf{y}; \mathcal{D}_k^{\text{train}}\right) + c\|\mathbf{y}\|^2, \tag{14}$$

where, $\mathcal{D}_k^{\text{train}} := \{(\tilde{\mathbf{a}}_{k,i}, \tilde{b}_{k,i})\}_{i=1}^{m}$ is the noisy training data set and $\mathcal{D}_k^{\text{val}} := \{(\mathbf{a}_{k,i}, b_{k,i})\}_{i=1}^{n}$ denotes a clean validation data set. Here, $\sigma(\mathbf{x}_{k,i})$ is the sigmoid function, $l_k()$ is the cross entropy loss at each node $k \in N$ and $c$ is the regularization constant. The lower level problem aims to find the optimal model parameters $\mathbf{y}$ that minimizes the weighted average of the loss function (with regularization) on the noisy dataset $\mathcal{D}_k^{\text{train}}$ for a fixed set of coefficients $\sigma(\mathbf{x}_{k,i})$, $k \in \mathcal{G}$. On the other hand, the upper-level optimization problem aims to find the coefficients $\sigma(\mathbf{x}_{k,i})$ by minimizing the validation loss learned on the clean dataset $\mathcal{D}_k^{\text{val}}$ for $k \in \mathcal{G}$.

**Learnable Regularization:** The goal of learnable regularization is to learn the optimal regularization coefficients for newsgroup dataset (Chen et al., 2023; Liu et al., 2022a). The federated bilevel optimization problem for this task can be posed as:

$$\min_{\mathbf{x}} \frac{1}{Gm} \sum_{k=1}^{G} \sum_{i=1}^{m} l_k \left( \mathbf{y}^*(\mathbf{x}); \mathcal{D}_k^{\text{val}} \right),$$

subject to

$$\mathbf{y}^*(\mathbf{x}) \in \arg\min_{\mathbf{y}} \frac{1}{Gm} \sum_{k=1}^{G} \sum_{i=1}^{n} l_k \left( \mathbf{y}, \mathcal{D}_k^{\text{train}} \right) + \|W_{\mathbf{x}_k} \mathbf{y}\|^2,$$

where, $\mathcal{D}_k^{\text{train}} \coloneqq \{(\tilde{\mathbf{a}}_{k,i}, \tilde{b}_{k,i})\}_{i=1}^{m}$ the training data set and $\mathcal{D}_k^{\text{val}} \coloneqq \{(\mathbf{a}_{k,i}, b_{k,i})\}_{i=1}^{n}$ denotes the validation data set. The lower level function minimizes the loss on the training dataset across the good nodes by finding the optimal model parameters for a given regularization coefficient. The upper level objective finds the optimal regualrization coefficient by minimizing the validation dataset of good nodes.

We train a linear model on the MNIST dataset with $N = 16$ nodes. We divide the dataset into equal parts among $G$ good nodes, and ensured that the data distribution is heterogeneous. Also, we consider Byzantine nodes which have access to the entire dataset. We have used the bucketing algorithm followed by geometric median aggregator. Next, we present the four different attacks that we consider:

**Bit Flipping (BF):** In this attack, the Byzantine nodes change the sign of the gradient of the penalty function computed using the entire data set, i.e., it sends $-\nabla h_\lambda(\mathbf{x}, \mathbf{y})$ aiming to cancel the effect of the gradients shared by all the good nodes. See (Karimireddy et al., 2020; Rammal et al., 2024) for more details.
**Label Flipping (LF):** Byzantine nodes modify the label of the MNIST dataset to $9 - \mathbf{z}$, $z \in \{0, 1, \ldots, 9\}$.
**Inner product Manipulation (IPM):** The byzantine nodes will send $\frac{\Delta}{G} \sum_{k \in \mathcal{G}} \nabla h_{\lambda,k}(\mathbf{x}, \mathbf{y})$, where $\Delta$ decides the intensity of the attack (Xie et al., 2020).
**A Little is Enough (ALIE):** The byzantine nodes send $\mu_{\mathcal{G}} - \nu\sigma_{\mathcal{G}}$ to the server, where $\mu_{\mathcal{G}}$ and $\sigma_{\mathcal{G}}$ are the mean and the standard deviations of the good nodes gradients, respectively. Here, $\nu$ dictates the intensity of the attack (Baruch et al., 2019).

Next, we present the experimental results for Rob-FedBOB algorithm and corroborate our theoretical findings made in this paper for $N = 16$ nodes:

**Effect of $\alpha$ and heterogeneity:** Figure 1 and 3 show the convergence rate of the proposed algorithm versus the communication rounds $R$ under different attacks for the application of the data hypercleaning and learnable regularization, respectively. We have chosen $B = 3$ and $B = 6$ for a total of $N = 16$ nodes which result in approximately $\alpha = 0.2$ and $\alpha = 0.4$, respectively. It is clear from the figure that the gradients does not converge to zero due to the presence of Byzantine nodes, as expected. The figures also demonstrate that the gradient converges to a constant that scales with $\alpha$. This corroborates with Theorem 1, where we have shown that $\frac{1}{R} \sum_{r=0}^{R-1} \|\nabla h_\lambda(\mathbf{x}_r, \mathbf{y}_r)\|^2$ increases as $\alpha$ increases because of $\alpha(\zeta_f^2 + \lambda^2 \zeta_g^2)$ term, provided $\lambda$ and the inter-client heterogeneity are fixed. Figures 2 and 4 show the plot of the constraint violation (in log scale) versus the number of communication rounds $R$ under different attacks for data hypercleaning and learnable regularization applications, respectively. In particular, the figure shows that by increasing $\alpha$ for a fixed $\lambda = 1$ results in a poor violation performance. This corroborates our theoretical findings (see Theorem 2), where we have proved that $\frac{\text{Viol}_R}{R}$ increases as $\alpha$ increases at the rate of $\alpha(\zeta_f^2 + \lambda^2 \zeta_g^2)$.

In this subsection, we use the same setting as explained in our experimental results section (see 4).
**Effect of $\lambda$:** Figures 5(a) and (b) show the plot of convergence rate (in log scale) and the constraint violation (in log scale) versus $R$ by varying $\lambda$ while fixing $\alpha = 0.2$. The heterogeneity in the data causes the Rob-FedBOB to convergence to a non-zero gradient and violation, as expected. Note that Theorem 1 shows that the average gradient $\|\nabla h_\lambda(\mathbf{x}_r, \mathbf{y}_r)\|^2$ scales with $\lambda$ as $\frac{\lambda^2}{R}$. On the contrary, Fig. 5(b) shows that larger $\lambda$ results in a better violation. This is in agreement with Theorem 2 where we have shown that $\text{Viol}_R$ decreases at the rate of $\frac{1}{\lambda^2}$.

**Effect of $T$:** Figure 6(a) and (b) show the effect of $T$ on the convergence and constraint violation of Rob-FedBOB algorithm. We get $\mathcal{O}(\frac{1}{R})$ rate when $T \geq \mathcal{O}(\log(R))$ rounds.

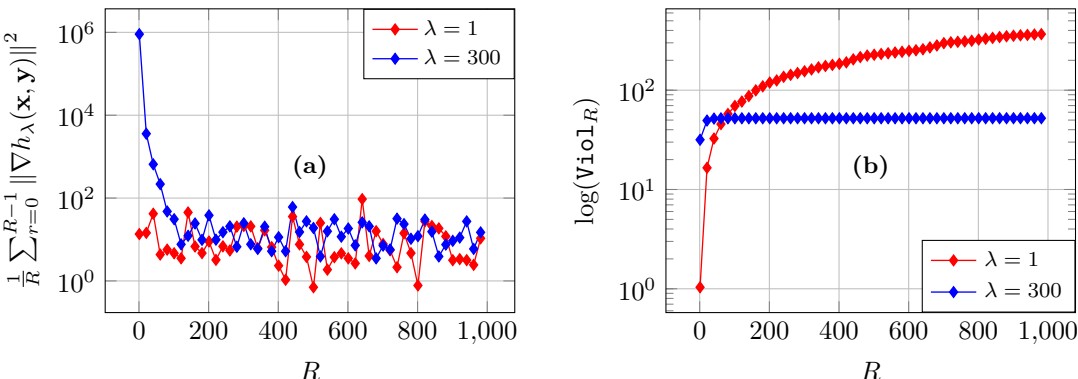

Figure 5: Effect of $\lambda$ on the convergence of Rob-FedBOB (see (a)) and violation (see (b)) under BF attack in the log scale for the data hyperclearning application.

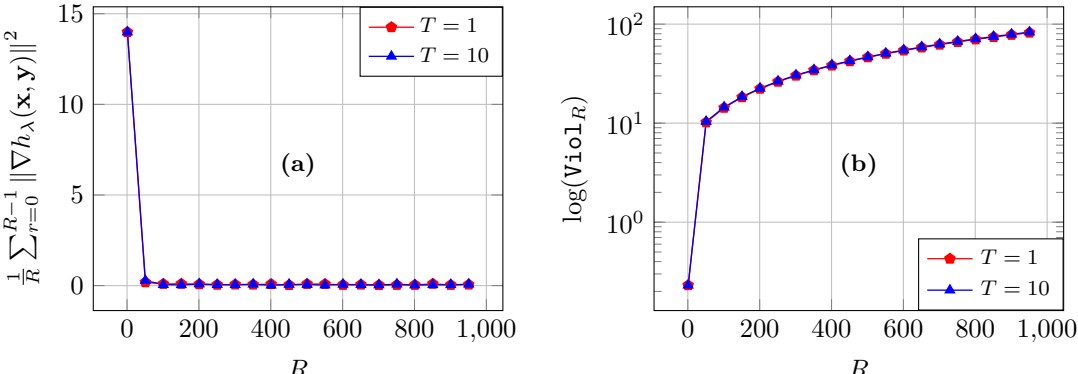

Figure 6: Effect of $T$ on the convergence of Rob-FedBOB (see (a)) and violation (see (b)) under LF attack in the log scale for the data hypercleaning application.

**Performance Comparison:** The Figure 7 shows the performance comparison between our proposed method (Rob-FedBOB) and Hessian-based federated bilevel optimization algorithm (BILANTINE) for the data hyper cleaning application. We have considered the bit flipping attack in both the cases. In Kwon et al. (2023b), the authors have demonstrated the superiority of the penalty based method over the second order (Hessian) method in the absence of Byzantine nodes in the centralized case. We on the other hand show superiority of our penalty method compared to BILA-TINE (second order method) in the FL setting in the presence of Byzantine nodes. As stated in Kwon et al. (2023b), the exact mathematical reason of why the penalty methods work better than the second order methods is not clear, and will be relagated to the future work.

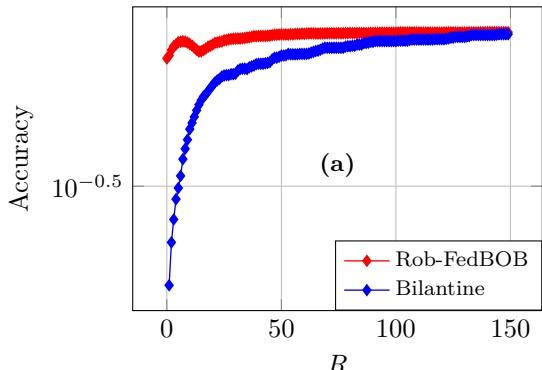

Figure 7: Performance comparison of Rob-FedBOB with BILATINE.

# 5 Conclusion

In this work, we considered the problem of bilevel optimization with Byzantine nodes where both upper and lower level objective functions are non-convex. Typically, bilevel problems are solved using an estimate of the hypergradient which requires second-order information about the lower level objective making it computationally expensive. Further, the presence of Byzantine nodes require the algorithm to be robust against attacks. We propose Rob-FedBOB algorithm, a fully first order byzantine robust federated algorithm that (i) does not require second-order information making it computationally efficient, (ii) handles non-convex lower level objective function satisfying PL-inequality and heterogeneous data, and (iii) aggregates the information from all the nodes in a robust manner making it resilient to Byzantine attacks. We prove theoretical performance of the proposed algorithm, and show that the rate at which the gradient and the average violation scale depends heavily on the fraction of the Byzantine nodes and the heterogeneity of the upper and lower level objective functions. In the absence of Byzantine nodes, we show that the gradient converges at the rate of $1/R$, where $R$ is the number of communication rounds while the violation can be made arbitrarily small. We have performed extensive experiments to corroborate our theoretical findings and to demonstrate that the complexity of Rob-FedBOB is very low.

**Acknowledgements:** This work is in part supported by MEiTY, India AI Mission (2024-2026), SERB-CRG (grant number: CRG/2021/007502) and SERB-MATRICS (grant number: MTR/2021/000575). The authors would like to thank the anonymous reviewers for their valuable comments.

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

## A    Appendix

### A.1   Useful Lemma

In this subsection, we prove a lemma that relates the gradient of the lower level functions with and without aggregation.

> **Lemma 6.** *Suppose the aggregator* `RAgg` *is* $(\alpha, c)$ *robust aggregator (see definition 1), then*
>
> $$\left\| \nabla_{\mathbf{y}} g_{ag} \left( \mathbf{x}_r, \mathbf{y}_{r,t} \right) - \nabla_{\mathbf{y}} g \left( \mathbf{x}_r, \mathbf{y}_{r,t} \right) \right\|^2 \quad \leq \quad 4 c \alpha \zeta_g^2.$$

*Proof:* It follows from the definition 1 of $(\alpha, c)$-Robust Aggregator that the above result can be shown provided we prove a bound on $\mathbb{E} \left\| \nabla_{\mathbf{y}} g_i \left( \mathbf{x}_r, \mathbf{y}_{r,t} \right) - \nabla_{\mathbf{y}} g_j \left( \mathbf{x}_r, \mathbf{y}_{r,t} \right) \right\|^2$, where the expectation is with respect to uniformly randomly chosen $i, j \in \mathcal{G}$. Adding and subtracting the term $\nabla_{\mathbf{y}} g \left( \mathbf{x}_r, \mathbf{y}_r \right)$ and using the inequality $\left\| a + b \right\|^2 \leq 2 \left\| a \right\|^2 + 2 \left\| b \right\|^2$, we get

$$
\begin{aligned}
\mathbb{E} \left\| \nabla_{\mathbf{y}} g_i \left( \mathbf{x}_r, \mathbf{y}_{r,t} \right) - \nabla_{\mathbf{y}} g_j \left( \mathbf{x}_r, \mathbf{y}_{r,t} \right) \right\|^2 \quad &\leq \quad 2 \mathbb{E}_{i \in \mathcal{G}} \left\| \nabla_{\mathbf{y}} g_i \left( \mathbf{x}_r, \mathbf{y}_r \right) - \nabla_{\mathbf{y}} g \left( \mathbf{x}_r, \mathbf{y}_{r,t} \right) \right\|^2 + \\
&\qquad 2 \mathbb{E}_{j \in \mathcal{G}} \left\| \nabla_{\mathbf{y}} g_j \left( \mathbf{x}_r, \mathbf{y}_r \right) - \nabla_{\mathbf{y}} g \left( \mathbf{x}_r, \mathbf{y}_{r,t} \right) \right\|^2 \\
&= \quad 4 \mathbb{E}_{i \in \mathcal{G}} \left\| \nabla_{\mathbf{y}} g_i \left( \mathbf{x}_r, \mathbf{y}_r \right) - \nabla_{\mathbf{y}} g \left( \mathbf{x}_r, \mathbf{y}_{r,t} \right) \right\|^2 . \quad (15)
\end{aligned}
$$

Using the inter-client heterogeneity in Assumption 3, the above is bounded as

$$\mathbb{E}_{i \in \mathcal{G}} \left\| \nabla_{\mathbf{y}} g_i \left( \mathbf{x}_r, \mathbf{y}_r \right) - \nabla_{\mathbf{y}} g \left( \mathbf{x}_r, \mathbf{y}_{r,t} \right) \right\|^2 \quad \leq \quad \zeta_g^2. \quad (16)$$

Using this in equation 15, we get

$$\mathbb{E} \left\| \nabla_{\mathbf{y}} g_i \left( \mathbf{x}_r, \mathbf{y}_{r,t} \right) - \nabla_{\mathbf{y}} g_j \left( \mathbf{x}_r, \mathbf{y}_{r,t} \right) \right\|^2 \quad \leq \quad 4 \zeta_g^2.$$

From the definition 1 of $(\alpha, c)$-Robust Aggregator, it follows that

$$\left\| \nabla_{\mathbf{y}} g_{\mathbf{ag}} \left( \mathbf{x}_r, \mathbf{y}_{r,t} \right) - \nabla_{\mathbf{y}} g \left( \mathbf{x}_r, \mathbf{y}_{r,t} \right) \right\|^2 \quad \leq \quad 4 c \alpha \zeta_g^2.$$

This completes the proof. $\qquad \square$

## B   Proof of Lemma 1

In this section, we prove a bound on $\left\| \nabla_{\mathbf{y}} h_{\lambda, \mathbf{ag}} \left( \mathbf{x}_r, \mathbf{y}_r \right) - \nabla_{\mathbf{y}} h_\lambda \left( \mathbf{x}_r, \mathbf{y}_r \right) \right\|^2$ to show robustness of the aggregator with respect to the penalty function. From Definition 1, we know that robustness can be proved by proving a bound on $\mathbb{E} \left\| \nabla_{\mathbf{y}} h_{\lambda, i} \left( \mathbf{x}_r, \mathbf{y}_r \right) - \nabla_{\mathbf{x}} h_{\lambda, j} \left( \mathbf{x}_r, \mathbf{y}_r \right) \right\|^2$, where expectation is with respect $i, j$ uniformly sampled from $\mathcal{G}$. Adding and subtracting the term $\nabla_{\mathbf{y}} h_\lambda \left( \mathbf{x}_r, \mathbf{y}_r \right)$ and using the inequality $\left\| a + b \right\|^2 \leq 2 \left\| a \right\|^2 + 2 \left\| b \right\|^2$, we get

$$
\begin{aligned}
\mathbb{E} \left\| \nabla_{\mathbf{y}} h_{\lambda, i} \left( \mathbf{x}_r, \mathbf{y}_r \right) - \nabla_{\mathbf{y}} h_{\lambda, j} \left( \mathbf{x}_r, \mathbf{y}_r \right) \right\|^2 \quad &\leq \quad 2 \mathbb{E}_{i \in \mathcal{G}} \left\| \nabla_{\mathbf{y}} h_{\lambda, i} \left( \mathbf{x}_r, \mathbf{y}_r \right) - \nabla_{\mathbf{y}} h_\lambda \left( \mathbf{x}_r, \mathbf{y}_r \right) \right\|^2 + \\
&\qquad 2 \mathbb{E}_{j \in \mathcal{G}} \left\| \nabla_{\mathbf{y}} h_{\lambda, j} \left( \mathbf{x}_r, \mathbf{y}_r \right) - \nabla_{\mathbf{y}} h_\lambda \left( \mathbf{x}_r, \mathbf{y}_r \right) \right\|^2 \\
&= \quad 4 \mathbb{E}_{i \in \mathcal{G}} \left\| \nabla_{\mathbf{y}} h_{\lambda, i} \left( \mathbf{x}_r, \mathbf{y}_r \right) - \nabla_{\mathbf{y}} h_\lambda \left( \mathbf{x}_r, \mathbf{y}_r \right) \right\|^2 . \quad (17)
\end{aligned}
$$

Consider the above equation, i.e., $\mathcal{A} := \mathbb{E}_{i \in \mathcal{G}} \left\| \nabla_{\mathbf{y}} h_{\lambda, i} \left( \mathbf{x}_r, \mathbf{y}_r \right) - \nabla_{\mathbf{y}} h_\lambda \left( \mathbf{x}_r, \mathbf{y}_r \right) \right\|^2$. Substituting $\nabla_{\mathbf{y}} h_{\lambda, i} \left( \mathbf{x}_r, \mathbf{y}_r \right) = \nabla_{\mathbf{y}} f_i \left( \mathbf{x}_r, \mathbf{y}_r \right) + \lambda \nabla_{\mathbf{y}} g_i \left( \mathbf{x}_r, \mathbf{y}_r \right)$ for all $i \in \mathcal{G}$ and $\nabla_{\mathbf{y}} h_\lambda \left( \mathbf{x}_r, \mathbf{y}_r \right) = \nabla_{\mathbf{y}} f \left( \mathbf{x}_r, \mathbf{y}_r \right) + \lambda \nabla_{\mathbf{y}} g \left( \mathbf{x}_r, \mathbf{y}_r \right)$, we get

$$
\begin{aligned}
\mathcal{A} \quad &= \quad \mathbb{E}_{i \in \mathcal{G}} \left\| \nabla_{\mathbf{y}} f_i \left( \mathbf{x}_r, \mathbf{y}_r \right) + \lambda \nabla_{\mathbf{y}} g_i \left( \mathbf{x}_r, \mathbf{y}_r \right) - \nabla_{\mathbf{y}} f \left( \mathbf{x}_r, \mathbf{y}_r \right) - \lambda \nabla_{\mathbf{y}} g \left( \mathbf{x}_r, \mathbf{y}_r \right) \right\|^2 \\
&\overset{(a)}{\leq} \quad 2 \mathbb{E}_{i \in \mathcal{G}} \left\| \nabla_{\mathbf{y}} f_i \left( \mathbf{x}_r, \mathbf{y}_r \right) - \nabla_{\mathbf{y}} f \left( \mathbf{x}_r, \mathbf{y}_r \right) \right\|^2 + 2 \mathbb{E}_{i \in \mathcal{G}} \lambda^2 \left\| \nabla_{\mathbf{y}} g_i \left( \mathbf{x}_r, \mathbf{y}_r \right) - \nabla_{\mathbf{y}} g \left( \mathbf{x}_r, \mathbf{y}_r \right) \right\|^2, \\
&\leq \quad 2 \zeta_f^2 + 2 \lambda^2 \zeta_g^2, \quad (18)
\end{aligned}
$$

where $(a)$ follows from the inequality, $\|a + b\|^2 \leq 2\|a\|^2 + 2\|b\|^2$, and the last inequality follows from the heterogeneity Assumption 3. Substituting the above in equation 17, we get

$$\mathbb{E}\|\nabla_{\mathbf{y}} h_{\lambda,i}(\mathbf{x}_r, \mathbf{y}_r) - \nabla_{\mathbf{y}} h_{\lambda,j}(\mathbf{x}_r, \mathbf{y}_r)\|^2 \leq 8\zeta_f^2 + 8\lambda^2\zeta_g^2.$$

From the definition 1 of $(\alpha, c)$-Robust Aggregator, it is clear that the output of robust aggregator satisfies

$$\left\|\nabla_{\mathbf{y}} h_{\lambda,\mathbf{ag}}(\mathbf{x}_r, \mathbf{y}_r) - \nabla_{\mathbf{y}} h_{\lambda}(\mathbf{x}_r, \mathbf{y}_r)\right\|^2 \leq c\alpha\left(8\zeta_f^2 + 8\lambda^2\zeta_g^2\right).$$

This completes the proof. $\qquad\square$

## C  Proof of Lemma 2

In this section, we prove a bound on $\left\|\nabla_{\mathbf{x}}\hat{h}_{\lambda,\mathbf{ag}}(\mathbf{x}_r, \mathbf{y}_r) - \nabla_{\mathbf{x}}\bar{\hat{h}}_{\lambda}(\mathbf{x}_r, \mathbf{y}_r)\right\|^2$. Towards this, we need to first prove a bound on (see Definition 1) $\mathbb{E}\left\|\nabla_{\mathbf{x}}\hat{h}_{\lambda,i}(\mathbf{x}_r, \mathbf{y}_r) - \nabla_{\mathbf{x}}\hat{h}_{\lambda,j}(\mathbf{x}_r, \mathbf{y}_r)\right\|^2$, where $i, j$ are uniformly chosen from $\mathcal{G}$. Adding and subtracting the term $\nabla_{\mathbf{x}} h_{\lambda}(\mathbf{x}_r, \mathbf{y}_r)$ to the above and using the inequality $\|a + b\|^2 \leq 2\|a\|^2 + 2\|b\|^2$, we get

$$
\begin{aligned}
\mathbb{E}\left\|\nabla_{\mathbf{x}}\hat{h}_{\lambda,i}(\mathbf{x}_r, \mathbf{y}_r) - \nabla_{\mathbf{x}}\hat{h}_{\lambda,j}(\mathbf{x}_r, \mathbf{y}_r)\right\|^2 &\leq 2\mathbb{E}_{i\in\mathcal{G}}\left\|\nabla_{\mathbf{x}}\hat{h}_{\lambda,i}(\mathbf{x}_r, \mathbf{y}_r) - \nabla_{\mathbf{x}} h_{\lambda}(\mathbf{x}_r, \mathbf{y}_r)\right\|^2 + \\
&\quad 2\mathbb{E}_{j\in\mathcal{G}}\left\|\nabla_{\mathbf{x}}\hat{h}_{\lambda,j}(\mathbf{x}_r, \mathbf{y}_r) - \nabla_{\mathbf{x}} h_{\lambda}(\mathbf{x}_r, \mathbf{y}_r)\right\|^2 \\
&= 4\mathbb{E}_{i\in\mathcal{G}}\left\|\nabla_{\mathbf{x}}\hat{h}_{\lambda,i}(\mathbf{x}_r, \mathbf{y}_r) - \nabla_{\mathbf{x}} h_{\lambda}(\mathbf{x}_r, \mathbf{y}_r)\right\|^2. \qquad(19)
\end{aligned}
$$

The following lemma is required to prove a bound on the above equation.

**Lemma 7.** *Under Assumption 3, the following bound is satisfied*

$$\mathbb{E}_{i\in\mathcal{G}}\|\nabla_{\mathbf{x}} h_{\lambda,i}(\mathbf{x}_r, \mathbf{y}_r) - \nabla_{\mathbf{x}} h_{\lambda,j}(\mathbf{x}_r, \mathbf{y}_r)\|^2 \leq 4\lambda^2 L_{g,\max}^2 d_{S(\mathbf{x}_r)}^2(\mathbf{y}_{r,T}) + 6\zeta_f^2 + 12\lambda^2\zeta_g^2. \qquad(20)$$

*Proof:* Towards bounding the above equation, let us add and subtract the term $\nabla_{\mathbf{x}}\bar{\hat{h}}_{\lambda}(\mathbf{x}_r, \mathbf{y}_r) := \frac{1}{G}\sum_{k\in\mathcal{G}}\hat{h}_{\lambda,j}(\mathbf{x}_r, \mathbf{y}_r)$ (see Lemma 2)

$$
\begin{aligned}
\mathbb{E}_{i\in\mathcal{G}}\left\|\nabla_{\mathbf{x}}\hat{h}_{\lambda,i}(\mathbf{x}_r, \mathbf{y}_r) - \nabla_{\mathbf{x}} h_{\lambda}(\mathbf{x}_r, \mathbf{y}_r)\right\|^2 &= \mathbb{E}_{i\in\mathcal{G}}\left\|\nabla_{\mathbf{x}}\hat{h}_{\lambda,i}(\mathbf{x}_r, \mathbf{y}_r) - \nabla_{\mathbf{x}}\bar{\hat{h}}_{\lambda}(\mathbf{x}_r, \mathbf{y}_r) + \right. \\
&\quad\left. \nabla_{\mathbf{x}}\bar{\hat{h}}_{\lambda}(\mathbf{x}_r, \mathbf{y}_r) - \nabla_{\mathbf{x}} h_{\lambda}(\mathbf{x}_r, \mathbf{y}_r)\right\|^2 \\
&\leq \underbrace{2\,\mathbb{E}_{i\in\mathcal{G}}\left\|\nabla_{\mathbf{x}}\hat{h}_{\lambda,i}(\mathbf{x}_r, \mathbf{y}_r) - \nabla_{\mathbf{x}}\bar{\hat{h}}_{\lambda}(\mathbf{x}_r, \mathbf{y}_r)\right\|^2}_{:=\mathcal{A}_1} + \\
&\quad \underbrace{2\left\|\nabla_{\mathbf{x}}\bar{\hat{h}}_{\lambda}(\mathbf{x}_r, \mathbf{y}_r) - \nabla_{\mathbf{x}} h_{\lambda}(\mathbf{x}_r, \mathbf{y}_r)\right\|^2}_{:=\mathcal{A}_2}, \qquad(21)
\end{aligned}
$$

where the above follows from the inequality $\|a + b\|^2 \leq 2\|a\|^2 + 2\|b\|^2$. Consider

$$\mathcal{A}_1 = \mathbb{E}_{i\in\mathcal{G}}\left\|\nabla_{\mathbf{x}}\hat{h}_{\lambda,i}(\mathbf{x}_r, \mathbf{y}_r) - \nabla_{\mathbf{x}}\bar{\hat{h}}_{\lambda}(\mathbf{x}_r, \mathbf{y}_r)\right\|^2.$$

Substituting $\nabla_{\mathbf{x}}\hat{h}_{\lambda,i}(\mathbf{x}_r, \mathbf{y}_r) = \nabla_{\mathbf{x}}f_i(\mathbf{x}_r, \mathbf{y}_r) + \lambda(\nabla_{\mathbf{x}}g_i(\mathbf{x}_r, \mathbf{y}_r) - \nabla_{\mathbf{x}}g_i(\mathbf{x}_r, \mathbf{y}_{r,T}))$ from equation 9, and $\nabla_{\mathbf{x}}\bar{\hat{h}}_{\lambda}(\mathbf{x}_r, \mathbf{y}_r) = \nabla_{\mathbf{x}}f(\mathbf{x}_r, \mathbf{y}_r) + \lambda(\nabla_{\mathbf{x}}g(\mathbf{x}_r, \mathbf{y}_r) - \nabla_{\mathbf{x}}g(\mathbf{x}_r, \mathbf{y}_{r,T}))$ in the above, we get

$$
\begin{aligned}
\mathcal{A}_1 &= \mathbb{E}_{i\in\mathcal{G}}\|\nabla_{\mathbf{x}}f_i(\mathbf{x}_r, \mathbf{y}_r) + \lambda(\nabla_{\mathbf{x}}g_i(\mathbf{x}_r, \mathbf{y}_r) - \nabla_{\mathbf{x}}g_i(\mathbf{x}_r, \mathbf{y}_{r,T})) - \nabla_{\mathbf{x}}f(\mathbf{x}_r, \mathbf{y}_r) - \\
&\quad \lambda(\nabla_{\mathbf{x}}g(\mathbf{x}_r, \mathbf{y}_r) - \nabla_{\mathbf{x}}g(\mathbf{x}_r, \mathbf{y}_{r,T}))\|^2 \\
&\overset{(a)}{\leq} 3\mathbb{E}_{i\in\mathcal{G}}\|\nabla_{\mathbf{x}}f_i(\mathbf{x}_r, \mathbf{y}_r) - \nabla_{\mathbf{x}}f(\mathbf{x}_r, \mathbf{y}_r)\|^2 + 3\mathbb{E}_{i\in\mathcal{G}}\lambda^2\|\nabla_{\mathbf{x}}g_i(\mathbf{x}_r, \mathbf{y}_r) - \nabla_{\mathbf{x}}g(\mathbf{x}_r, \mathbf{y}_r)\|^2 \\
&\quad + 3\mathbb{E}_{i\in\mathcal{G}}\lambda^2\|\nabla_{\mathbf{x}}g_i(\mathbf{x}_r, \mathbf{y}_{r,T}) - \nabla_{\mathbf{x}}g(\mathbf{x}_r, \mathbf{y}_{r,T})\|^2,
\end{aligned} \tag{22}
$$

where $(a)$ follows from the inequality, $\|a + b + c\|^2 \leq 3\|a\|^2 + 3\|b\|^2 + 3\|c\|^2$. Now, consider

$$
\begin{aligned}
\mathcal{A}_1 &\leq 3\mathbb{E}_i\|\nabla_{\mathbf{x}}f_i(\mathbf{x}_r, \mathbf{y}_r) - \nabla_{\mathbf{x}}f(\mathbf{x}_r, \mathbf{y}_r)\|^2 + 3\lambda^2\mathbb{E}_i\|\nabla_{\mathbf{x}}g_i(\mathbf{x}_r, \mathbf{y}_r) - \nabla_{\mathbf{x}}g(\mathbf{x}_r, \mathbf{y}_r)\|^2 \\
&\quad + 3\lambda^2\mathbb{E}_i\|\nabla_{\mathbf{x}}g_i(\mathbf{x}_r, \mathbf{y}_{r,T}) - \nabla_{\mathbf{x}}g(\mathbf{x}_r, \mathbf{y}_{r,T})\|^2 \\
&\leq 3\zeta_f^2 + 6\lambda^2\zeta_g^2,
\end{aligned} \tag{23}
$$

where the last inequality follows from the inter-client heterogeneity Assumption 3, i.e., $\mathbb{E}_{i\in\mathcal{G}}\|\nabla f_i(\mathbf{x}, \mathbf{y}) - \nabla f(\mathbf{x}, \mathbf{y})\|^2 \leq \zeta_f^2$ and $\mathbb{E}_{i\in\mathcal{G}}\|\nabla g_i(\mathbf{x}, \mathbf{y}) - \nabla g(\mathbf{x}, \mathbf{y})\|^2 \leq \zeta_g^2$. Let us consider the term $\mathcal{A}_2$ from equation 21

$$
\mathcal{A}_2 = \left\|\nabla_{\mathbf{x}}\bar{\hat{h}}_{\lambda}(\mathbf{x}_r, \mathbf{y}_r) - \nabla_{\mathbf{x}}h(\mathbf{x}_r, \mathbf{y}_r)\right\|^2.
$$

Substituting $\nabla_{\mathbf{x}}\bar{\hat{h}}_{\lambda}(\mathbf{x}_r, \mathbf{y}_r) = \nabla_{\mathbf{x}}f(\mathbf{x}_r, \mathbf{y}_r) + \lambda(\nabla_{\mathbf{x}}g(\mathbf{x}_r, \mathbf{y}_r) - \nabla_{\mathbf{x}}g(\mathbf{x}_r, \mathbf{y}_{r,T}))$ and $\nabla_{\mathbf{x}}h(\mathbf{x}_r, \mathbf{y}_r) = \nabla_{\mathbf{x}}f(\mathbf{x}_r, \mathbf{y}_r) - \lambda(\nabla_{\mathbf{x}}g(\mathbf{x}_r, \mathbf{y}_r) - \nabla_{\mathbf{x}}v(\mathbf{x}_r))$ in the above, we get

$$
\mathcal{A}_2 \leq \|\nabla_{\mathbf{x}}f(\mathbf{x}_r, \mathbf{y}_r) + \lambda(\nabla_{\mathbf{x}}g(\mathbf{x}_r, \mathbf{y}_r) - \nabla_{\mathbf{x}}g(\mathbf{x}_r, \mathbf{y}_{r,T})) - \nabla_{\mathbf{x}}f(\mathbf{x}_r, \mathbf{y}_r) - \lambda(\nabla_{\mathbf{x}}g(\mathbf{x}_r, \mathbf{y}_r) - \nabla_{\mathbf{x}}v(\mathbf{x}_r))\|^2.
$$

Simplifying further, we get

$$
\mathcal{A}_2 \leq \lambda^2\|\nabla_{\mathbf{x}}v(\mathbf{x}_r) - \nabla_{\mathbf{x}}g(\mathbf{x}_r, \mathbf{y}_{r,T})\|^2.
$$

Using Jensen's inequality, we get

$$
\begin{aligned}
\mathcal{A}_2 &\leq \frac{\lambda^2}{G}\sum_{k=1}^{G}\|\nabla_{\mathbf{x}}v_k(\mathbf{x}_r) - \nabla_{\mathbf{x}}g_k(\mathbf{x}_r, \mathbf{y}_{r,T})\|^2 \\
&\overset{(a)}{\leq} \frac{\lambda^2}{G}\sum_{k=1}^{G}L_{g,k}^2 d_{S(\mathbf{x}_r)}^2(\mathbf{y}_{r,T}) \\
&\overset{(b)}{\leq} \lambda^2 L_{g,\max}^2 d_{S(\mathbf{x}_r)}^2(\mathbf{y}_{r,T}),
\end{aligned} \tag{24}
$$

where $(a)$ follows from the Assumption 1, and $L_{g,\max}^2 := \max_{k\in\mathcal{G}}L_{g,k}^2$. Substituting equation 23, equation 24 in equation 21, we get

$$
\mathbb{E}_{i\in\mathcal{G}}\left\|\nabla_{\mathbf{x}}\hat{h}_{\lambda,i}(\mathbf{x}_r, \mathbf{y}_r) - \nabla_{\mathbf{x}}h_{\lambda}(\mathbf{x}_r, \mathbf{y}_r)\right\|^2 \leq 2\lambda^2 L_{g,\max}^2 d_{S(\mathbf{x}_r)}^2(\mathbf{y}_{r,T}) + 6\zeta_f^2 + 12\lambda^2\zeta_g^2. \tag{25}
$$

Using the results from equation 25 in equation 19, we get the desired result. This completes the proof of the Lemma. $\qquad\square$

To complete the proof of Lemma 2, we substitute the result from Lemma 5 in equation 20 to get

$$
\begin{aligned}
\mathbb{E}_{i\in\mathcal{G}}\|\nabla_{\mathbf{x}}h_{\lambda,i}(\mathbf{x}_r, \mathbf{y}_r) - \nabla_{\mathbf{x}}h_{\lambda,j}(\mathbf{x}_r, \mathbf{y}_r)\|^2 &\leq \frac{8\lambda^2 L_{g,\max}^2 l_{g,\max}^2}{\mu_g^2}\left(1 - \frac{\gamma\mu_g}{2}\right)^T + \frac{16\lambda^2 L_{g,\max}^2 c\alpha\zeta_g^2}{\mu_g} \\
&\quad + 6\zeta_f^2 + 12\lambda^2\zeta_g^2.
\end{aligned}
$$

Using the definition 1 of $(\alpha, c)$-robust aggregator, the output of the robust aggregator satisfies the following

$$\left\|\nabla_{\mathbf{x}} \hat{h}_{\lambda,\mathbf{ag}}\left(\mathbf{x}_r, \mathbf{y}_r\right) - \nabla_{\mathbf{x}} \bar{\hat{h}}_{\lambda}\left(\mathbf{x}_r, \mathbf{y}_r\right)\right\|^2 \leq c\alpha\left(\frac{8\lambda^2 L_{g,\max}^2 l_{g,\max}^2}{\mu_g^2}\left(1 - \frac{\gamma\mu_g}{2}\right)^T + \frac{16\lambda^2 L_{g,\max}^2 c\alpha\zeta_g^2}{\mu_g}\right.$$
$$\left. + 6\zeta_f^2 + 12\lambda^2\zeta_g^2\right). \tag{26}$$

This completes the proof. $\qquad\square$

## D   Proof of Lemma 3

Suppose $(\mathbf{x}_r, \mathbf{y}_r)$ is an approximate stationary point of `FedBOB Penalty Problem`, i.e.,

$$\|\nabla_{\mathbf{y}} h_{\lambda}(\mathbf{x}_r, \mathbf{y}_r)\| = \|\nabla_{\mathbf{y}} f\left(\mathbf{x}_r, \mathbf{y}_r\right) + \lambda\nabla_{\mathbf{y}} g\left(\mathbf{x}_r, \mathbf{y}_r\right)\| \leq \psi_r, \tag{27}$$

where $\psi_r > 0$ is the approximation error. Then, for some $\mathbf{v}_r$ such that $\|\mathbf{v}_r\| \leq \psi_r$, the above can be written as

$$\nabla_{\mathbf{y}} f\left(\mathbf{x}_r, \mathbf{y}_r\right) = -\lambda\nabla_{\mathbf{y}} g\left(\mathbf{x}_r, \mathbf{y}_r\right) + \mathbf{v}_r.$$

Rearranging the above, taking norms on both sides and applying triangle's inequality, we get

$$\lambda\|\nabla_{\mathbf{y}} g\left(\mathbf{x}_r, \mathbf{y}_r\right)\| \leq \|\nabla_{\mathbf{y}} f\left(\mathbf{x}_r, \mathbf{y}_r\right)\| + \psi_r.$$

Squaring on both the sides, we get

$$\|\nabla_{\mathbf{y}} g\left(\mathbf{x}_r, \mathbf{y}_r\right)\|^2 \overset{(a)}{\leq} \frac{2}{\lambda^2}\left(\|\nabla_{\mathbf{y}} f\left(\mathbf{x}_r, \mathbf{y}_r\right)\|^2 + \psi_r^2\right)$$
$$\overset{(b)}{\leq} \frac{2}{\lambda^2}\left(\frac{1}{G}\sum_{k=0}^{G}\|\nabla_{\mathbf{y}} f_k\left(\mathbf{x}_r, \mathbf{y}_r\right)\|^2 + \psi_r^2\right)$$
$$\overset{(c)}{\leq} \frac{2\left(l_f^2 + \psi_r^2\right)}{\lambda^2}, \tag{28}$$

where $(a)$ follows from the inequality $\|a + b\|^2 \leq 2\|a\|^2 + 2\|b\|^2$, $(b)$ uses the Jensen's inequality, and $(c)$ follows from Assumption 1 i.e., $\|\nabla_{\mathbf{y}} f_k\left(\mathbf{x}, \mathbf{y}\right)\|^2 \leq l_f^2$ for all $k \in \mathcal{G}$. From Assumption 2, we know that

$$g\left(\mathbf{x}_r, \mathbf{y}_r\right) - g_t\left(\mathbf{x}_r, \mathbf{y}^*(\mathbf{x}_r)\right) \leq \frac{1}{\mu_g}\|\nabla_{\mathbf{y}} g\left(\mathbf{x}_r, \mathbf{y}_r\right)\|^2. \tag{29}$$

Substituting equation 28 in equation 29, we get

$$p\left(\mathbf{x}_r, \mathbf{y}_r\right) \coloneqq g\left(\mathbf{x}_r, \mathbf{y}_r\right) - g\left(\mathbf{x}_r, \mathbf{y}^*(\mathbf{x}_r)\right) \leq \frac{2l_f^2}{\mu_g\lambda^2} + \frac{2\psi_r^2}{\mu_g\lambda^2}.$$

Summing from $r = 0$ to $R - 1$, and dividing by $R$, we get

$$\frac{\mathtt{Viol}_R}{R} \coloneqq \frac{1}{R}\sum_{r=0}^{R-1} p\left(\mathbf{x}_r, \mathbf{y}_r\right) \leq \frac{2l_f^2}{\mu_g\lambda^2} + \frac{2}{\mu_g\lambda^2} \times \frac{1}{R}\sum_{r=0}^{R-1}\psi_r^2. \tag{30}$$

Note that the result in Lemma 3 can be obtained by choosing $(\mathbf{x}_r, \mathbf{y}_r) = (\mathbf{x}, \mathbf{y})$ for all $r$ satisfying $\|\nabla_{\mathbf{y}} h_{\lambda}(\mathbf{x}, \mathbf{y})\| \leq \psi$. This completes the proof. $\qquad\square$

## E   Proof of Lemma 4 and Lemma 5

In this section, we prove smoothness of the penalty function in Lemma 4 and a bound on $d_{S(\mathbf{x}_r)}^2(\mathbf{y}_{r,T})$ in Lemma E.0.2. Recall that $\mathbf{y}_{r,T}$ is the output of the Algorithm 1 in step 12 which is used as a proxy for $\mathbf{y}^* \in \arg\min_{\mathbf{y}} g(\mathbf{x}_r, \mathbf{y})$. First, we state the proof of Lemma 4.

### E.0.1 Proof of Lemma 4

Recall from equation 3 that the penalty function $h_\lambda(\mathbf{x}, \mathbf{y}) := f(\mathbf{x}, \mathbf{y}) + \lambda p(\mathbf{x}, \mathbf{y})$. Consider the following

$$
\begin{aligned}
\|\nabla_{\mathbf{y}} h_\lambda(\mathbf{x}, \mathbf{y}) - \nabla_{\mathbf{y}} h_\lambda(\mathbf{x}, \mathbf{y}')\| &= \|\nabla_{\mathbf{y}} f(\mathbf{x}, \mathbf{y}) + \lambda \nabla_{\mathbf{y}} p(\mathbf{x}, \mathbf{y}) - \nabla_{\mathbf{y}} f(\mathbf{x}, \mathbf{y}') - \lambda \nabla_{\mathbf{y}} p(\mathbf{x}, \mathbf{y}')\| \\
&\leq \|\nabla_{\mathbf{y}} f(\mathbf{x}, \mathbf{y}) - \nabla_{\mathbf{y}} f(\mathbf{x}, \mathbf{y}')\| + \lambda \|\nabla_{\mathbf{y}} p(\mathbf{x}, \mathbf{y}) - \nabla_{\mathbf{y}} p(\mathbf{x}, \mathbf{y}')\|.
\end{aligned}
$$

The above inequality follows from the triangle's inequality i.e., $\|a + b\| \leq \|a\| + \|b\|$. From equation 2, we know that $p(\mathbf{x}, \mathbf{y}') = g(\mathbf{x}, \mathbf{y}') - g(\mathbf{x}, \mathbf{y}^*)$, where $\mathbf{y}^* \in \arg\min_{\mathbf{y}} g(\mathbf{x}, \mathbf{y})$. Using the fact that $\nabla_{\mathbf{y}} g(\mathbf{x}, \mathbf{y}^*) = 0$ in the above, we get

$$
\begin{aligned}
\|\nabla h_\lambda(\mathbf{x}, \mathbf{y}) - \nabla h_\lambda(\mathbf{x}, \mathbf{y}')\| &= \|\nabla_{\mathbf{y}} f(\mathbf{x}, \mathbf{y}) - \nabla_{\mathbf{y}} f(\mathbf{x}, \mathbf{y}')\| + \lambda \|\nabla_{\mathbf{y}} g(\mathbf{x}, \mathbf{y}) - \nabla_{\mathbf{y}} g(\mathbf{x}, \mathbf{y}')\| \\
&\leq \left\|\frac{1}{G}\sum_{k=1}^{G}(\nabla_{\mathbf{y}} f_k(\mathbf{x}, \mathbf{y}) - \nabla_{\mathbf{y}} f_k(\mathbf{x}, \mathbf{y}'))\right\| + \lambda \left\|\frac{1}{G}\sum_{k=1}^{G}(\nabla_{\mathbf{y}} g_k(\mathbf{x}, \mathbf{y}) - \nabla_{\mathbf{y}} g_k(\mathbf{x}, \mathbf{y}'))\right\| \\
&\overset{(a)}{\leq} \frac{1}{G}\sum_{k=1}^{G} L_{f,k}\|\mathbf{y} - \mathbf{y}'\| + \frac{\lambda}{G}\sum_{k=1}^{G} L_{g,k}\|\mathbf{y} - \mathbf{y}'\| \\
&\overset{(b)}{\leq} (L_{f,\max} + \lambda L_{g,\max})\|\mathbf{y} - \mathbf{y}'\|. \\
&\overset{(c)}{\leq} L_h\|\mathbf{y} - \mathbf{y}'\|,
\end{aligned}
$$

where $(a)$ follows from the triangle's inequality and Assumption 1 and $(b)$ uses the fact that $L_{f,\max} := \max_{k \in \mathcal{G}} L_{f,k}$ and $L_{g,\max} := \max_{k \in \mathcal{G}} L_{g,k}$, and $(c)$ results from $L_h := L_{f,\max} + \lambda L_{g,\max}$. This completes the proof. $\square$

### E.0.2 Proof of Lemma 5

First, note that in Lemma 5, we have used $\mathbf{y}_{r,T} = \mathbf{y}_{r,T}$. Using smoothness property from the Assumption 1, we have

$$
g(\mathbf{x}_r, \mathbf{y}_{r,t+1}) \leq g(\mathbf{x}_r, \mathbf{y}_{r,t}) + \langle \mathbf{y}_{r,t+1} - \mathbf{y}_{r,t}, \nabla_{\mathbf{y}} g(\mathbf{x}_r, \mathbf{y}_{r,t})\rangle + \frac{L_g}{2}\|\mathbf{y}_{r,t+1} - \mathbf{y}_{r,t}\|^2.
$$

Substituting the update in 6 i.e., $\mathbf{y}_{r,t+1} = \mathbf{y}_{r,t} - \gamma \nabla_{\mathbf{y}} g_{\mathbf{ag}}(\mathbf{x}_r, \mathbf{y}_{r,t})$ in the above, we have

$$
g(\mathbf{x}_r, \mathbf{y}_{r,t+1}) \leq g(\mathbf{x}_r, \mathbf{y}_{r,t}) - \gamma \langle \nabla_{\mathbf{y}} g_{\mathbf{ag}}(\mathbf{x}_r, \mathbf{y}_{r,t}), \nabla_{\mathbf{y}} g(\mathbf{x}_r, \mathbf{y}_{r,t})\rangle + \frac{\gamma^2 L_g}{2}\|\nabla_{\mathbf{y}} g_{\mathbf{ag}}(\mathbf{x}_r, \mathbf{y}_{r,t})\|^2.
$$

Using the identity $\langle a, b\rangle = \frac{1}{2}\|a\|^2 + \frac{1}{2}\|b\|^2 - \frac{1}{2}\|a - b\|^2$, we get

$$
\begin{aligned}
g(\mathbf{x}_r, \mathbf{y}_{r,t+1}) \leq{}& g(\mathbf{x}_r, \mathbf{y}_{r,t}) - \frac{\gamma}{2}\|\nabla_{\mathbf{y}} g_{\mathbf{ag}}(\mathbf{x}_r, \mathbf{y}_{r,t})\|^2 - \frac{\gamma}{2}\|\nabla_{\mathbf{y}} g(\mathbf{x}_r, \mathbf{y}_{r,t})\|^2 \\
&+ \frac{\gamma}{2}\|\nabla_{\mathbf{y}} g_{\mathbf{ag}}(\mathbf{x}_r, \mathbf{y}_{r,t}) - \nabla_{\mathbf{y}} g(\mathbf{x}_r, \mathbf{y}_{r,t})\|^2 + \frac{\gamma^2 L_g}{2}\|\nabla_{\mathbf{y}} g_{\mathbf{ag}}(\mathbf{x}_r, \mathbf{y}_{r,t})\|^2.
\end{aligned} \tag{31}
$$

Choosing $\gamma \leq \frac{1}{L_g}$ in the above, and ignoring the negative term, we get

$$
g(\mathbf{x}_r, \mathbf{y}_{r,t+1}) \leq g(\mathbf{x}_r, \mathbf{y}_{r,t}) - \frac{\gamma}{2}\|\nabla_{\mathbf{y}} g(\mathbf{x}_r, \mathbf{y}_{r,t})\|^2 + \frac{\gamma}{2}\|\nabla_{\mathbf{y}} g_{\mathbf{ag}}(\mathbf{x}_r, \mathbf{y}_{r,t}) - \nabla_{\mathbf{y}} g(\mathbf{x}_r, \mathbf{y}_{r,t})\|^2. \tag{32}
$$

Using the result of Lemma 6, the third term in the above is bounded as

$$
\|\nabla_{\mathbf{y}} g_{\mathbf{ag}}(\mathbf{x}_r, \mathbf{y}_{r,t}) - \nabla_{\mathbf{y}} g(\mathbf{x}_r, \mathbf{y}_{r,t})\|^2 \leq 4c\alpha\zeta_g^2. \tag{33}
$$

Using the PL-inequality in Assumption 2 i.e., $\|\nabla g(\mathbf{x}_r, \mathbf{y}_{r,t})\|^2 \geq \mu_g(g(\mathbf{x}_r, \mathbf{y}_{r,t}) - v(\mathbf{x}_r))$ and substituting equation 33 in equation 32, we get

$$
g(\mathbf{x}_r, \mathbf{y}_{r,t+1}) \leq g(\mathbf{x}_r, \mathbf{y}_{r,t}) - \frac{\gamma\mu_g}{2}(g(\mathbf{x}_r, \mathbf{y}_{r,t}) - v(\mathbf{x}_r)) + 2\gamma c\alpha\zeta_g^2
$$

Adding and subtracting the term $v(\mathbf{x}_r)$, we have

$$g(\mathbf{x}_r, \mathbf{y}_{r,t+1}) - v(\mathbf{x}_r) \leq \left(1 - \frac{\gamma \mu_g}{2}\right)(g(\mathbf{x}_r, \mathbf{y}_{r,t}) - v(\mathbf{x}_r)) + 2\gamma c\alpha \zeta_g^2.$$

Applying recursion, we have

$$g(\mathbf{x}_r, \mathbf{y}_{r,T}) - v(\mathbf{x}_r) \leq \left(1 - \frac{\gamma \mu_g}{2}\right)^T(g(\mathbf{x}_r, \mathbf{y}_r) - v(\mathbf{x}_r)) + \sum_{t=0}^{T-1}\left(1 - \frac{\gamma \mu_g}{2}\right)^t 2\gamma c\alpha \zeta_g^2.$$

It is important to note that the PL-inequality implies quadratic growth condition, i.e., $g(\mathbf{x}_r, \mathbf{y}_{r,T}) - v(\mathbf{x}_r) \geq \frac{\mu_g}{2}d_{S(\mathbf{x}_r)}^2(\mathbf{y}_{r,T})$ for some $\mu_g > 0$. Since $g()$ satisfies the PL-inequality, invoking the quadratic growth condition, we get

$$d_{S(\mathbf{x}_r)}^2(\mathbf{y}_{r,T}) \leq \frac{2}{\mu_g}\left(1 - \frac{\gamma \mu_g}{2}\right)^T(g(\mathbf{x}_r, \mathbf{y}_r) - v(\mathbf{x}_r)) + \sum_{t=0}^{T-1}\left(1 - \frac{\gamma \mu_g}{2}\right)^t 2\gamma c\alpha \zeta_g^2.$$

Next, we can further bound the above using the fact that the lower level function satisfies the PL-inequality (see Assumption 2):

$$d_{S(\mathbf{x}_r)}^2(\mathbf{y}_{r,T}) \leq \frac{2}{\mu_g^2}\left(1 - \frac{\gamma \mu_g}{2}\right)^T \|\nabla g(\mathbf{x}_r, \mathbf{y}_r)\|^2 + \sum_{t=0}^{T-1}\left(1 - \frac{\gamma \mu_g}{2}\right)^t 2\gamma c\alpha \zeta_g^2. \tag{34}$$

Using Assumption 1, i.e., $\|\nabla g_k(\mathbf{x}_r, \mathbf{y}_r)\|^2 \leq l_{g,k}^2$, we have $\|\nabla g(\mathbf{x}_r, \mathbf{y}_r)\|^2 \leq \frac{1}{G}\sum_{k=1}^{G}\|\nabla g_k(\mathbf{x}_r, \mathbf{y}_r)\|^2 \leq l_{g,\max}^2$. Using this in the above results in

$$d_{S(\mathbf{x}_r)}^2(\mathbf{y}_{r,T}) \leq \frac{2l_{g,\max}^2}{\mu_g^2}\left(1 - \frac{\gamma \mu_g}{2}\right)^T + \sum_{t=0}^{T-1}\left(1 - \frac{\gamma \mu_g}{2}\right)^t 2\gamma c\alpha \zeta_g^2. \tag{35}$$

The above is further bounded using geometric series as

$$d_{S(\mathbf{x}_r)}^2(\mathbf{y}_{r,T}) \leq \frac{2l_{g,\max}^2}{\mu_g^2}\left(1 - \frac{\gamma \mu_g}{2}\right)^T + \frac{4c\alpha \zeta_g^2}{\mu_g}. \tag{36}$$

This completes the proof. $\qquad\square$

## F  Proof of Theorem 1

First, we state and prove the following Lemma, which is also useful while proving the violation bound.

**Lemma 8.** *Suppose assumptions 1-3 hold, then for the aggregator* `RAgg`*, Algorithm 1 achieves the following bound*

$$\min\left\{\frac{\beta}{2}, \frac{\eta}{2}\right\}\|\nabla h_\lambda(\mathbf{x}_r, \mathbf{y}_r)\|^2 \leq (f(\mathbf{x}_r, \mathbf{y}_r) - f(\mathbf{x}_{r+1}, \mathbf{y}_{r+1})) + \lambda(g(\mathbf{x}_r, \mathbf{y}_r) - g(\mathbf{x}_{r+1}, \mathbf{y}_{r+1})) -$$

$$\lambda(v(\mathbf{x}_{r+1}) - v(\mathbf{x}_r))) + \beta c\alpha \delta^2 + \frac{\eta}{2}c\alpha \rho^2 + \frac{2\beta \lambda^2 L_{g,\max}^2 l_{g,\max}^2}{\mu_g^2}\left(1 - \frac{\gamma \mu_g}{2}\right)^T$$

$$+ \frac{2\beta \lambda^2 L_{g,\max}^2 c\alpha \zeta_g^2}{\mu_g}. \tag{37}$$

*for* $\beta \leq 1/L_h$ *and* $\eta \leq 1/L_h$, *and for any* $T \geq 1$.

*Proof:* Using the smoothness result from Lemma 4, we can write

$$
\begin{aligned}
h_\lambda\left(\mathbf{x}_{r+1}, \mathbf{y}_{r+1}\right) &\leq h_\lambda\left(\mathbf{x}_r, \mathbf{y}_r\right)+\left\langle\mathbf{x}_{r+1}-\mathbf{x}_r, \nabla_\mathbf{x} h_\lambda\left(\mathbf{x}_r, \mathbf{y}_r\right)\right\rangle+\left\langle\mathbf{y}_{r+1}-\mathbf{y}_r, \nabla_\mathbf{y} h_\lambda\left(\mathbf{x}_r, \mathbf{y}_r\right)\right\rangle \\
&+\frac{L_h}{2}\left[\left\|\mathbf{x}_{r+1}-\mathbf{x}_r\right\|^2+\left\|\mathbf{y}_{r+1}-\mathbf{y}_r\right\|^2\right].
\end{aligned}
\tag{38}
$$

Substituting for the update $\mathbf{x}_{r+1}-\mathbf{x}_r=-\beta \nabla_\mathbf{x} \hat{h}_{\lambda, \mathbf{ag}}\left(\mathbf{x}_r, \mathbf{y}_r\right)$ and $\mathbf{y}_{r+1}-\mathbf{y}_r=-\eta \nabla_\mathbf{y} h_{\lambda, \mathbf{ag}}\left(\mathbf{x}_r, \mathbf{y}_r\right)$ from equation 8 and equation 7, we get

$$
\begin{aligned}
h_\lambda\left(\mathbf{x}_{r+1}, \mathbf{y}_{r+1}\right) &\leq h_\lambda\left(\mathbf{x}_r, \mathbf{y}_r\right)-\beta\left\langle\nabla_\mathbf{x} \hat{h}_{\lambda, \mathbf{ag}}\left(\mathbf{x}_r, \mathbf{y}_r\right), \nabla_\mathbf{x} h_\lambda\left(\mathbf{x}_r, \mathbf{y}_r\right)\right\rangle-\eta\left\langle\nabla_\mathbf{y} h_{\lambda, \mathbf{ag}}\left(\mathbf{x}_r, \mathbf{y}_r\right), \nabla_\mathbf{y} h_\lambda\left(\mathbf{x}_r, \mathbf{y}_r\right)\right\rangle \\
&+\frac{\beta^2 L_h}{2}\left\|\nabla_\mathbf{x} \hat{h}_{\lambda, \mathbf{ag}}\left(\mathbf{x}_r, \mathbf{y}_r\right)\right\|^2+\frac{\eta^2 L_h}{2}\left\|\nabla_\mathbf{y} h_{\lambda, \mathbf{ag}}\left(\mathbf{x}_r, \mathbf{y}_r\right)\right\|^2.
\end{aligned}
$$

Using the identity $\langle a, b\rangle=\frac{1}{2}\|a\|^2+\frac{1}{2}\|b\|^2-\frac{1}{2}\|a-b\|^2$, and adding and subtracting the term $\nabla_\mathbf{x} \bar{\hat{h}}_\lambda\left(\mathbf{x}_r, \mathbf{y}_r\right)$, we have

$$
\begin{aligned}
h_\lambda\left(\mathbf{x}_{r+1}, \mathbf{y}_{r+1}\right) &\leq h_\lambda\left(\mathbf{x}_r, \mathbf{y}_r\right)-\frac{\beta}{2}\left\|\nabla_\mathbf{x} \hat{h}_{\lambda, \mathbf{ag}}\left(\mathbf{x}_r, \mathbf{y}_r\right)\right\|^2-\frac{\beta}{2}\left\|\nabla_\mathbf{x} h_\lambda\left(\mathbf{x}_r, \mathbf{y}_r\right)\right\|^2 \\
&+\frac{\beta}{2}\left\|\nabla_\mathbf{x} \hat{h}_{\lambda, \mathbf{ag}}\left(\mathbf{x}_r, \mathbf{y}_r\right)-\nabla_\mathbf{x} \bar{\hat{h}}_\lambda\left(\mathbf{x}_r, \mathbf{y}_r\right)+\nabla_\mathbf{x} \bar{\hat{h}}_\lambda\left(\mathbf{x}_r, \mathbf{y}_r\right)-\nabla_\mathbf{x} h_\lambda\left(\mathbf{x}_r, \mathbf{y}_r\right)\right\|^2 \\
&-\frac{\eta}{2}\left\|\nabla_\mathbf{y} h_{\lambda, \mathbf{ag}}\left(\mathbf{x}_r, \mathbf{y}_r\right)\right\|^2-\frac{\eta}{2}\left\|\nabla_\mathbf{y} h_\lambda\left(\mathbf{x}_r, \mathbf{y}_r\right)\right\|^2+\frac{\eta}{2}\left\|\nabla_\mathbf{y} h_{\lambda, \mathbf{ag}}\left(\mathbf{x}_r, \mathbf{y}_r\right)-\nabla_\mathbf{y} h_\lambda\left(\mathbf{x}_r, \mathbf{y}_r\right)\right\|^2 \\
&+\frac{\beta^2 L_h}{2}\left\|\nabla_\mathbf{x} \hat{h}_{\lambda, \mathbf{ag}}\left(\mathbf{x}_r, \mathbf{y}_r\right)\right\|^2+\frac{\eta^2 L_h}{2}\left\|\nabla_\mathbf{y} h_{\lambda, \mathbf{ag}}\left(\mathbf{x}_r, \mathbf{y}_r\right)\right\|^2.
\end{aligned}
$$

Combining the common terms, choosing $\eta \leq \frac{1}{L_h}$ and $\beta \leq \frac{1}{L_h}$, and applying the inequality $\|a+b\|^2 \leq 2\|a\|^2+2\|b\|^2$, we get

$$
\begin{aligned}
h_\lambda\left(\mathbf{x}_{r+1}, \mathbf{y}_{r+1}\right) &\leq h_\lambda\left(\mathbf{x}_r, \mathbf{y}_r\right)-\frac{\beta}{2}\left\|\nabla_\mathbf{x} h_\lambda\left(\mathbf{x}_r, \mathbf{y}_r\right)\right\|^2+\beta\left\|\nabla_\mathbf{x} \hat{h}_{\lambda, \mathbf{ag}}\left(\mathbf{x}_r, \mathbf{y}_r\right)-\nabla_\mathbf{x} \bar{\hat{h}}_\lambda\left(\mathbf{x}_r, \mathbf{y}_r\right)\right\|^2 \\
&+\beta\left\|\nabla_\mathbf{x} \bar{\hat{h}}_\lambda\left(\mathbf{x}_r, \mathbf{y}_r\right)-\nabla_\mathbf{x} h_\lambda\left(\mathbf{x}_r, \mathbf{y}_r\right)\right\|^2-\frac{\eta}{2}\left\|\nabla_\mathbf{y} h_\lambda\left(\mathbf{x}_r, \mathbf{y}_r\right)\right\|^2+ \\
&\frac{\eta}{2}\left\|\nabla_\mathbf{y} h_{\lambda, \mathbf{ag}}\left(\mathbf{x}_r, \mathbf{y}_r\right)-\nabla_\mathbf{y} h_\lambda\left(\mathbf{x}_r, \mathbf{y}_r\right)\right\|^2.
\end{aligned}
\tag{39}
$$

Using $\nabla_\mathbf{x} \hat{h}_{\lambda, k}\left(\mathbf{x}_r, \mathbf{y}_r\right):=\nabla_\mathbf{x} f_k\left(\mathbf{x}_r, \mathbf{y}_r\right)+\lambda\left(\nabla_\mathbf{x} g_k\left(\mathbf{x}_r, \mathbf{y}_r\right)-\nabla_\mathbf{x} g_k\left(\mathbf{x}_r, \mathbf{y}_{r, T}\right)\right)$ in the fourth term above, we get

$$
\begin{aligned}
\left\|\nabla_\mathbf{x} \bar{\hat{h}}_\lambda\left(\mathbf{x}_r, \mathbf{y}_r\right)-\nabla_\mathbf{x} h_\lambda\left(\mathbf{x}_r, \mathbf{y}_r\right)\right\|^2 &\leq \left\|\nabla_\mathbf{x} f\left(\mathbf{x}_r, \mathbf{y}_r\right)+\lambda\left(\nabla_\mathbf{x} g\left(\mathbf{x}_r, \mathbf{y}_r\right)-\frac{1}{G} \sum_{k=1}^G \nabla_\mathbf{x} g_k\left(\mathbf{x}_r, \mathbf{y}_{r, T}\right)\right)\right. \\
&\left.-\nabla_\mathbf{x} f\left(\mathbf{x}_r, \mathbf{y}_r\right)-\lambda\left(\nabla_\mathbf{x} g\left(\mathbf{x}_r, \mathbf{y}_r\right)-\frac{1}{G} \sum_{k=1}^G \nabla_\mathbf{x} v_k\left(\mathbf{x}_r\right)\right)\right\|^2.
\end{aligned}
$$

Simplifying the above results in

$$
\left\|\nabla_\mathbf{x} \bar{\hat{h}}_\lambda\left(\mathbf{x}_r, \mathbf{y}_r\right)-\nabla_\mathbf{x} h_\lambda\left(\mathbf{x}_r, \mathbf{y}_r\right)\right\|^2 \leq \lambda^2\left\|\frac{1}{G} \sum_{k=1}^G \nabla_\mathbf{x} v_k\left(\mathbf{x}_r\right)-\frac{1}{G} \sum_{k=1}^G \nabla_\mathbf{x} g_k\left(\mathbf{x}_r, \mathbf{y}_{r, T}\right)\right\|^2.
$$

Using Jensen's inequality, the above is further bounded as

$$
\begin{aligned}
\left\|\nabla_\mathbf{x} \bar{\hat{h}}_\lambda\left(\mathbf{x}_r, \mathbf{y}_r\right)-\nabla_\mathbf{x} h_\lambda\left(\mathbf{x}_r, \mathbf{y}_r\right)\right\|^2 &\leq \frac{\lambda^2}{G} \sum_{k=1}^G\left\|\nabla_\mathbf{x} v_k\left(\mathbf{x}_r\right)-\nabla_\mathbf{x} g_k\left(\mathbf{x}_r, \mathbf{y}_{r, T}\right)\right\|^2 \\
&\overset{(a)}{\leq} \frac{\lambda^2}{G} \sum_{k=1}^G L_{g, k}^2 d_{S\left(\mathbf{x}_r\right)}^2\left(\mathbf{y}_{r, T}\right) \\
&\overset{(b)}{\leq} \lambda^2 L_{g, \max}^2 d_{S\left(\mathbf{x}_r\right)}^2\left(\mathbf{y}_{r, T}\right),
\end{aligned}
$$

where $(a)$ follows from the Assumption 1 and $(b)$ uses the fact that $L_{g,\max}^2 := \max_{k \in \mathcal{G}} L_{g,k}^2$. Now using the result from Lemma 5 in equation 40, we get

$$\left\| \nabla_{\mathbf{x}} \bar{\hat{h}}_\lambda (\mathbf{x}_r, \mathbf{y}_r) - \nabla_{\mathbf{x}} h_\lambda (\mathbf{x}_r, \mathbf{y}_r) \right\|^2 \leq \frac{2\lambda^2 L_{g,\max}^2 l_{g,\max}^2}{\mu_g^2} \left(1 - \frac{\gamma \mu_g}{2}\right)^T + \frac{4\lambda^2 L_{g,\max}^2 c\alpha \zeta_g^2}{\mu_g}. \tag{40}$$

From Lemma 2, the third term in equation 39 can be bounded as

$$\left\| \nabla_{\mathbf{x}} \hat{h}_{\lambda,\mathbf{ag}} (\mathbf{x}_r, \mathbf{y}_r) - \nabla_{\mathbf{x}} \bar{\hat{h}}_\lambda (\mathbf{x}_r, \mathbf{y}_r) \right\|^2 \leq c\alpha \delta^2, \tag{41}$$

where $\delta^2 := \frac{8\lambda^2 L_{g,\max}^2 l_{g,\max}^2}{\mu_g^2} \left(1 - \frac{\gamma \mu_g}{2}\right)^T + \frac{16\lambda^2 L_{g,\max}^2 c\alpha \zeta_g^2}{\mu_g} + 6\zeta_f^2 + 12\lambda^2 \zeta_g^2$. Similarly, from Lemma 1, the last term in equation 39 is bounded as

$$\left\| \nabla_{\mathbf{y}} h_{\lambda,\mathbf{ag}} (\mathbf{x}_r, \mathbf{y}_r) - \nabla_{\mathbf{y}} h_\lambda (\mathbf{x}_r, \mathbf{y}_r) \right\|^2 \leq c\alpha \rho^2. \tag{42}$$

where $\rho^2 := c\alpha \left(8\zeta_f^2 + 8\lambda^2 \zeta_g^2\right)$. Substituting equation 40, equation 41 and equation 42 in equation 39, we get

$$\begin{aligned} h_\lambda (\mathbf{x}_{r+1}, \mathbf{y}_{r+1}) &\leq h_\lambda (\mathbf{x}_r, \mathbf{y}_r) - \frac{\beta}{2} \|\nabla_{\mathbf{x}} h_\lambda (\mathbf{x}_r, \mathbf{y}_r)\|^2 - \frac{\eta}{2} \|\nabla_{\mathbf{y}} h_\lambda (\mathbf{x}_r, \mathbf{y}_r)\|^2 + \beta c\alpha \delta^2 + \frac{\eta}{2} c\alpha \rho^2 \\ &\quad + \frac{2\beta \lambda^2 L_{g,\max}^2 l_{g,\max}^2}{\mu_g^2} \left(1 - \frac{\gamma \mu_g}{2}\right)^T + \frac{2\beta \lambda^2 L_{g,\max}^2 c\alpha \zeta_g^2}{\mu_g}. \end{aligned}$$

Rearranging and using the fact that $h_\lambda (\mathbf{x}, \mathbf{y}) = f (\mathbf{x}, \mathbf{y}) + \lambda (g (\mathbf{x}, \mathbf{y}) - v (\mathbf{x}))$, the above becomes

$$\begin{aligned} \frac{\beta}{2} \|\nabla_{\mathbf{x}} h_\lambda (\mathbf{x}_r, \mathbf{y}_r)\|^2 + \frac{\eta}{2} \|\nabla_{\mathbf{y}} h_\lambda (\mathbf{x}_r, \mathbf{y}_r)\|^2 &\leq f (\mathbf{x}_r, \mathbf{y}_r) + \lambda (g (\mathbf{x}_r, \mathbf{y}_r) - v (\mathbf{x}_r)) \\ &\quad - f (\mathbf{x}_{r+1}, \mathbf{y}_{r+1}) - \lambda (g (\mathbf{x}_{r+1}, \mathbf{y}_{r+1}) - v (\mathbf{x}_{r+1})) \\ &\quad + \beta c\alpha \delta^2 + \frac{\eta}{2} c\alpha \rho^2 + \frac{2\beta \lambda^2 L_{g,\max}^2 l_{g,\max}^2}{\mu_g^2} \left(1 - \frac{\gamma \mu_g}{2}\right)^T \\ &\quad + \frac{2\beta \lambda^2 L_{g,\max}^2 c\alpha \zeta_g^2}{\mu_g}. \end{aligned} \tag{43}$$

Using the fact that, $\min \left\{\frac{\beta}{2}, \frac{\eta}{2}\right\} \|\nabla h_\lambda (\mathbf{x}_r, \mathbf{y}_r)\|^2 \leq \frac{\beta}{2} \|\nabla_{\mathbf{x}} h_\lambda (\mathbf{x}_r, \mathbf{y}_r)\|^2 + \frac{\eta}{2} \|\nabla_{\mathbf{y}} h_\lambda (\mathbf{x}_r, \mathbf{y}_r)\|^2$ and rearranging results in the bound of Lemma 8. This completes the proof of the lemma. $\qquad \square$

**Completing the Proof of Theorem 1:** Now, summing both sides of the the bound of Lemma 8 results in

$$\begin{aligned} \min \left\{\frac{\beta}{2}, \frac{\eta}{2}\right\} \sum_{r=0}^{R-1} \|\nabla h_\lambda (\mathbf{x}_r, \mathbf{y}_r)\|^2 &\leq (f (\mathbf{x}_0, \mathbf{y}_0) - f (\mathbf{x}_R, \mathbf{y}_R)) + \lambda (g (\mathbf{x}_0, \mathbf{y}_0) - g (\mathbf{x}_R, \mathbf{y}_R)) - \\ &\quad \lambda (v (\mathbf{x}_R)) - v (\mathbf{x}_0)) + \beta R c\alpha \delta^2 + \frac{\eta}{2} R c\alpha \rho^2 + \\ &\quad \frac{2\beta R \lambda^2 L_{g,\max}^2 l_{g,\max}^2}{\mu_g^2} \left(1 - \frac{\gamma \mu_g}{2}\right)^T + \frac{2\beta R \lambda^2 L_{g,\max}^2 c\alpha \zeta_g^2}{\mu_g}. \end{aligned} \tag{44}$$

Since we need $\eta, \beta \leq 1/L_h$, we assume that both $\eta$ and $\beta$ are of the same order. This results in

$$\begin{aligned} \frac{1}{2R} \sum_{r=0}^{R-1} \|\nabla h_\lambda (\mathbf{x}_r, \mathbf{y}_r)\|^2 &\leq \frac{(f (\mathbf{x}_0, \mathbf{y}_0) - f (\mathbf{x}_R, \mathbf{y}_R))}{\eta R} + \frac{\lambda (g (\mathbf{x}_0, \mathbf{y}_0) - g (\mathbf{x}_R, \mathbf{y}_R))}{\eta R} - \\ &\quad \frac{\lambda (v (\mathbf{x}_R)) - v (\mathbf{x}_0))}{\eta R} + c\alpha \delta^2 + \frac{1}{2} c\alpha \rho^2 + \frac{2\lambda^2 L_{g,\max}^2 l_{g,\max}^2}{\mu_g^2} \left(1 - \frac{\gamma \mu_g}{2}\right)^T \\ &\quad + \frac{2\lambda^2 L_{g,\max}^2 c\alpha \zeta_g^2}{\mu_g}. \end{aligned} \tag{45}$$

We wish the second last term on the right hand side in the above equation to satisfy

$$\frac{2\lambda^2 L_{g,\max}^2 l_{g,\max}^2}{\mu_g^2}\left(1-\frac{\gamma\mu_g}{2}\right)^T \leq \frac{1}{R}. \tag{46}$$

The above condition is achieved by choosing $T \geq \frac{2}{\gamma\mu_g}\log\left(\frac{2R\lambda^2 L_{g,\max}^2 l_{g,\max}^2}{\mu_g^2}\right)$; this makes the communication complexity slighter higher than the conventional second order methods. This will be relaxed in the next result that we state. Substituting equation 46 in equation 45, we get

$$
\begin{aligned}
\frac{1}{2R}\sum_{r=0}^{R-1}\|\nabla h_\lambda(\mathbf{x}_r,\mathbf{y}_r)\|^2 \leq{}& \frac{(f(\mathbf{x}_0,\mathbf{y}_0)-f(\mathbf{x}_R,\mathbf{y}_R))}{\eta R}+\frac{\lambda(g(\mathbf{x}_0,\mathbf{y}_0)-g(\mathbf{x}_R,\mathbf{y}_R))}{\eta R}-\\
& \frac{\lambda(v(\mathbf{x}_R))-v(\mathbf{x}_0))}{\eta R}+c\alpha\delta^2+\frac{1}{2}c\alpha\rho^2+\frac{1}{R}+\frac{2\lambda^2 L_{g,\max}^2 c\alpha\zeta_g^2}{\mu_g}.
\end{aligned}
$$

Now, substituting for $\delta^2$ and $\rho^2$, we get

$$
\begin{aligned}
\frac{1}{2R}\sum_{r=0}^{R-1}\|\nabla h_\lambda(\mathbf{x}_r,\mathbf{y}_r)\|^2 \leq{}& \frac{(f(\mathbf{x}_0,\mathbf{y}_0)-f(\mathbf{x}_R,\mathbf{y}_R))}{\eta R}+\frac{\lambda(g(\mathbf{x}_0,\mathbf{y}_0)-g(\mathbf{x}_R,\mathbf{y}_R))}{\eta R}-\frac{\lambda(v(\mathbf{x}_R))-v(\mathbf{x}_0))}{\eta R}+\\
& c\alpha\left(\frac{8\lambda^2 L_{g,\max}^2 l_{g,\max}^2}{\mu_g^2}\left(1-\frac{\gamma\mu_g}{2}\right)^T+\frac{8\lambda^2 L_{g,\max}^2 c\alpha\zeta_g^2}{\mu_g}+6\zeta_f^2+12\lambda^2\zeta_g^2\right)\\
& +\frac{1}{2}c\alpha\left(8\zeta_f^2+8\lambda^2\zeta_g^2\right)+\frac{1}{R}+\frac{2\lambda^2 L_{g,\max}^2 c\alpha\zeta_g^2}{\mu_g}.
\end{aligned}
$$

Suppose, we choose $T \geq \frac{2}{\gamma\mu_g}\log\left(\frac{8R\lambda^2 L_{g,\max}^2 l_{g,\max}^2}{\mu_g^2}\right)$, the above is further bounded as

$$
\begin{aligned}
\frac{1}{2R}\sum_{r=0}^{R-1}\|\nabla h_\lambda(\mathbf{x}_r,\mathbf{y}_r)\|^2 \leq{}& \frac{(f(\mathbf{x}_0,\mathbf{y}_0)-f(\mathbf{x}_R,\mathbf{y}_R))}{\eta R}+\frac{\lambda(g(\mathbf{x}_0,\mathbf{y}_0)-g(\mathbf{x}_R,\mathbf{y}_R))}{\eta R}+\\
& \frac{\lambda(v(\mathbf{x}_0))-v(\mathbf{x}_R))}{\eta R}+\frac{c\alpha}{R}+\frac{8\lambda^2 L_{g,\max}^2 c^2\alpha^2\zeta_g^2}{\mu_g}+6c\alpha\zeta_f^2+12c\alpha\lambda^2\zeta_g^2\\
& +4c\alpha\zeta_f^2+4c\alpha\lambda^2\zeta_g^2+\frac{1}{R}+\frac{2\lambda^2 L_{g,\max}^2 c\alpha\zeta_g^2}{\mu_g}.
\end{aligned}
$$

Rearranging the above, multiplying by 2 on both sides, and using only those terms that depend on $\lambda$ and $R$, we get the desired order result of the theorem. This completes the proof. $\qquad\square$

## G Proof of Theorem 2

In order to prove the Theorem, we need a bound on the stationary point of the penalty function (see Lemma 3. More specifically, we need a result of the form $\|\nabla h_\lambda(\mathbf{x}_r,\mathbf{y}_r)\|^2 \leq \psi_r^2$. Towards this, consider equation 37 of Lemma 8

$$
\begin{aligned}
\min\left\{\frac{\beta}{2},\frac{\eta}{2}\right\}\|\nabla h_\lambda(\mathbf{x}_r,\mathbf{y}_r)\|^2 \leq{}& (f(\mathbf{x}_r,\mathbf{y}_r)-f(\mathbf{x}_{r+1},\mathbf{y}_{r+1}))+\lambda(g(\mathbf{x}_r,\mathbf{y}_r)-g(\mathbf{x}_{r+1},\mathbf{y}_{r+1}))-\\
& \lambda(v(\mathbf{x}_{r+1})-v(\mathbf{x}_r)))+\beta c\alpha\delta^2+\frac{\eta}{2}c\alpha\rho^2+\\
& \frac{2\beta\lambda^2 L_{g,\max}^2 l_{g,\max}^2}{\mu_g^2}\left(1-\frac{\gamma\mu_g}{2}\right)^T+\frac{2\beta\lambda^2 L_{g,\max}^2 c\alpha\zeta_g^2}{\mu_g}. \tag{47}
\end{aligned}
$$

Dividing by $\lambda^2$ and using the fact that $\min\left\{\frac{\beta}{2}, \frac{\eta}{2}\right\} = \frac{1}{2L_h} = \frac{1}{2(L_{f,\max}+\lambda L_{g,\max})}$ and further rearranging, we get

$$
\begin{aligned}
\frac{1}{\lambda^2}\left\|\nabla h_\lambda\left(\mathbf{x}_r, \mathbf{y}_r\right)\right\|^2 \leq\ & \frac{2L_{f,\max}\left(f\left(\mathbf{x}_r, \mathbf{y}_r\right)-f\left(\mathbf{x}_{r+1}, \mathbf{y}_{r+1}\right)\right)}{\lambda^2}+\frac{2L_{g,\max}\left(f\left(\mathbf{x}_r, \mathbf{y}_r\right)-f\left(\mathbf{x}_{r+1}, \mathbf{y}_{r+1}\right)\right)}{\lambda}+ \\
& \frac{2L_{f,\max}\left(g\left(\mathbf{x}_r, \mathbf{y}_r\right)-g\left(\mathbf{x}_{r+1}, \mathbf{y}_{r+1}\right)\right)}{\lambda}+2L_{g,\max}\left(g\left(\mathbf{x}_r, \mathbf{y}_r\right)-g\left(\mathbf{x}_{r+1}, \mathbf{y}_{r+1}\right)\right)- \\
& \frac{2L_{f,\max}\left(v\left(\mathbf{x}_{r+1}\right)-v\left(\mathbf{x}_r\right)\right)}{\lambda}-2L_{g,\max}\left(v\left(\mathbf{x}_{r+1}\right)-v\left(\mathbf{x}_r\right)\right)+2c\alpha\delta^2 \\
& +c\alpha\rho^2+\frac{2\lambda^2 L_{g,\max}^2 l_{g,\max}^2}{\lambda^2\mu_g^2}\left(1-\frac{\gamma\mu_g}{2}\right)^T+\frac{2\lambda^2 L_{g,\max}^2 c\alpha\zeta_g^2}{\lambda^2\mu_g}.
\end{aligned}
$$

Choosing $T \geq \frac{2}{\gamma\mu_g}\log\left(\frac{2\lambda^2 L_{g,\max}^2 l_{g,\max}^2}{\mu_g^2}\right)$, we get

$$
\begin{aligned}
\frac{1}{\lambda^2}\left\|\nabla h_\lambda\left(\mathbf{x}_r, \mathbf{y}_r\right)\right\|^2 \leq\ & \frac{2L_{f,\max}\left(f\left(\mathbf{x}_r, \mathbf{y}_r\right)-f\left(\mathbf{x}_{r+1}, \mathbf{y}_{r+1}\right)\right)}{\lambda^2}+\frac{2L_{g,\max}\left(f\left(\mathbf{x}_r, \mathbf{y}_r\right)-f\left(\mathbf{x}_{r+1}, \mathbf{y}_{r+1}\right)\right)}{\lambda}+ \\
& \frac{2L_{f,\max}\left(g\left(\mathbf{x}_r, \mathbf{y}_r\right)-g\left(\mathbf{x}_{r+1}, \mathbf{y}_{r+1}\right)\right)}{\lambda}+2L_{g,\max}\left(g\left(\mathbf{x}_r, \mathbf{y}_r\right)-g\left(\mathbf{x}_{r+1}, \mathbf{y}_{r+1}\right)\right)- \\
& \frac{2L_{f,\max}\left(v\left(\mathbf{x}_{r+1}\right)-v\left(\mathbf{x}_r\right)\right)}{\lambda}-2L_{g,\max}\left(v\left(\mathbf{x}_{r+1}\right)-v\left(\mathbf{x}_r\right)\right)+2c\alpha\delta^2 \\
& +c\alpha\rho^2+\frac{1}{\lambda^2 R}+\frac{2\lambda^2 L_{g,\max}^2 c\alpha\zeta_g^2}{\lambda^2\mu_g}.
\end{aligned}
$$

Summing from $r=0$ to $R-1$ and simplifying, we get

$$
\begin{aligned}
\frac{1}{\lambda^2}\sum_{r=0}^{R-1}\left\|\nabla h_\lambda\left(\mathbf{x}_r, \mathbf{y}_r\right)\right\|^2 \leq\ & \sum_{r=0}^{R-1}\frac{2L_{f,\max}\left(f\left(\mathbf{x}_r, \mathbf{y}_r\right)-f\left(\mathbf{x}_{r+1}, \mathbf{y}_{r+1}\right)\right)}{\lambda^2}+\sum_{r=0}^{R-1}\frac{2L_{g,\max}\left(f\left(\mathbf{x}_r, \mathbf{y}_r\right)-f\left(\mathbf{x}_{r+1}, \mathbf{y}_{r+1}\right)\right)}{\lambda} \\
& +\sum_{r=0}^{R-1}\frac{2L_{f,\max}\left(g\left(\mathbf{x}_r, \mathbf{y}_r\right)-g\left(\mathbf{x}_{r+1}, \mathbf{y}_{r+1}\right)\right)}{\lambda}+2L_{g,\max}\sum_{r=0}^{R-1}\left(g\left(\mathbf{x}_r, \mathbf{y}_r\right)-g\left(\mathbf{x}_{r+1}, \mathbf{y}_{r+1}\right)\right) \\
& -\sum_{r=0}^{R-1}\frac{2L_{f,\max}\left(v\left(\mathbf{x}_{r+1}\right)-v\left(\mathbf{x}_r\right)\right)}{\lambda}-2L_{g,\max}\sum_{r=0}^{R-1}\left(v\left(\mathbf{x}_{r+1}\right)-v\left(\mathbf{x}_r\right)\right) \\
& +\sum_{r=0}^{R-1}\frac{2c\alpha\delta^2}{\lambda^2}+\sum_{r=0}^{R-1}\frac{c\alpha\rho^2}{\lambda^2}+\sum_{r=0}^{R-1}\frac{1}{\lambda^2 R}+\sum_{r=0}^{R-1}\frac{2\lambda^2 L_{g,\max}^2 c\alpha\zeta_g^2}{\lambda^2\mu_g}.
\end{aligned}
\tag{48}
$$

Using the telescopic sum, we get $\sum_{r=0}^{R-1}\left(f\left(\mathbf{x}_r, \mathbf{y}_r\right)-f\left(\mathbf{x}_{r+1}, \mathbf{y}_{r+1}\right)\right) = f\left(\mathbf{x}_0, \mathbf{y}_0\right)-f\left(\mathbf{x}_R, \mathbf{y}_R\right)$, $\sum_{r=0}^{R-1}\left(g\left(\mathbf{x}_r, \mathbf{y}_r\right)-g\left(\mathbf{x}_{r+1}, \mathbf{y}_{r+1}\right)\right) = g\left(\mathbf{x}_0, \mathbf{y}_0\right)-g\left(\mathbf{x}_R, \mathbf{y}_R\right)$ and $\sum_{r=0}^{R-1}\left(v\left(\mathbf{x}_{r+1}\right)-v\left(\mathbf{x}_r\right)\right) = v\left(\mathbf{x}_R\right)-v\left(\mathbf{x}_0\right)$. Using these results in equation 48, we get

$$
\begin{aligned}
\sum_{r=0}^{R-1}\frac{1}{\lambda^2}\left\|\nabla h_\lambda\left(\mathbf{x}_r, \mathbf{y}_r\right)\right\|^2 \leq\ & \left(\frac{2L_{f,\max}}{\lambda^2}+\frac{2L_{g,\max}}{\lambda}\right)\left(f\left(\mathbf{x}_0, \mathbf{y}_0\right)-f\left(\mathbf{x}_R, \mathbf{y}_R\right)\right)+ \\
& \left(\frac{2L_{f,\max}}{\lambda}+2L_{g,\max}\right)\left(g\left(\mathbf{x}_0, \mathbf{y}_0\right)-g\left(\mathbf{x}_R, \mathbf{y}_R\right)\right)+ \\
& \left(\frac{2L_{f,\max}}{\lambda}+2L_{g,\max}\right)\left(v\left(\mathbf{x}_0\right)-v\left(\mathbf{x}_R\right)\right)+\frac{2Rc\alpha\delta^2}{\lambda^2}+\frac{Rc\alpha\rho^2}{\lambda^2}+ \\
& \frac{1}{\lambda^2}+\frac{2R\lambda^2 L_{g,\max}^2 c\alpha\zeta_g^2}{\lambda^2\mu_g}.
\end{aligned}
\tag{49}
$$

From Lemma 3, we know that

$$
\mathtt{Viol}_R \leq \frac{2Rl_f^2}{\mu_g\lambda^2}+\frac{2}{\mu_g\lambda^2}\sum_{r=0}^{R-1}\psi_r^2.
\tag{50}
$$

Substituting equation 49 in equation 50, we get

$$
\begin{aligned}
\texttt{Viol}_R \ \leq \ & \frac{2Rl_f^2}{\mu_g \lambda^2} + \frac{2}{\mu_g}\left(\frac{2L_{f,\max}}{\lambda^2} + \frac{2L_{g,\max}}{\lambda}\right)\left(f\left(\mathbf{x}_0,\mathbf{y}_0\right) - f\left(\mathbf{x}_R,\mathbf{y}_R\right)\right) + \\
& \frac{2}{\mu_g}\left(\frac{2L_{f,\max}}{\lambda} + 2L_{g,\max}\right)\left(g\left(\mathbf{x}_0,\mathbf{y}_0\right) - g\left(\mathbf{x}_R,\mathbf{y}_R\right)\right) + \\
& \frac{2}{\mu_g}\left(\frac{2L_{f,\max}}{\lambda} + 2L_{g,\max}\right)\left(v\left(\mathbf{x}_0\right) - v\left(\mathbf{x}_R\right)\right) + \\
& \frac{4Rc\alpha\delta^2}{\mu_g \lambda^2} + \frac{2Rc\alpha\rho^2}{\mu_g \lambda^2} + \frac{1}{\lambda^2} + \frac{2R\lambda^2 L_{g,\max}^2 c\alpha\zeta_g^2}{\lambda^2 \mu_g}.
\end{aligned} \tag{51}
$$

Now substituting for $\delta^2$ and $\rho^2$, we get

$$
\begin{aligned}
\texttt{Viol}_R \ \leq \ & \frac{2Rl_f^2}{\mu_g \lambda^2} + \frac{2}{\mu_g}\left(\frac{2L_{f,\max}}{\lambda^2} + \frac{2L_{g,\max}}{\lambda}\right)\left(f\left(\mathbf{x}_0,\mathbf{y}_0\right) - f\left(\mathbf{x}_R,\mathbf{y}_R\right)\right) + \\
& \frac{2}{\mu_g}\left(\frac{2L_{f,\max}}{\lambda} + 2L_{g,\max}\right)\left(g\left(\mathbf{x}_0,\mathbf{y}_0\right) - g\left(\mathbf{x}_R,\mathbf{y}_R\right)\right) + \\
& \frac{2}{\mu_g}\left(\frac{2L_{f,\max}}{\lambda} + 2L_{g,\max}\right)\left(v\left(\mathbf{x}_0\right) - v\left(\mathbf{x}_R\right)\right) + \\
& \frac{4Rc\alpha}{\mu_g \lambda^2}\left(\frac{8\lambda^2 L_{g,\max}^2 l_{g,\max}^2}{\mu_g^2}\left(1 - \frac{\eta\mu_g}{2}\right)^T + \frac{8\lambda^2 L_{g,\max}^2 c\alpha\zeta_g^2}{\mu_g} + 6\zeta_f^2 + 12\lambda^2\zeta_g^2\right) + \\
& \frac{2Rc\alpha}{\mu_g \lambda^2}\left(8\zeta_f^2 + 8\lambda^2\zeta_g^2\right) + \frac{1}{\lambda^2} + \frac{2R\lambda^2 L_{g,\max}^2 c\alpha\zeta_g^2}{\lambda^2 \mu_g}.
\end{aligned}
$$

Now, choosing $T \geq \frac{2}{\eta\mu_g}\log\left(\frac{8R\lambda^2 L_{g,\max}^2 l_{g,\max}^2}{\mu_g^2}\right)$, we get

$$
\begin{aligned}
\texttt{Viol}_R \ \leq \ & \frac{2Rl_f^2}{\mu_g \lambda^2} + \frac{2}{\mu_g}\left(\frac{2L_{f,\max}}{\lambda^2} + \frac{2L_{g,\max}}{\lambda}\right)\left(f\left(\mathbf{x}_0,\mathbf{y}_0\right) - f\left(\mathbf{x}_R,\mathbf{y}_R\right)\right) + \\
& \frac{2}{\mu_g}\left(\frac{2L_{f,\max}}{\lambda} + 2L_{g,\max}\right)\left(g\left(\mathbf{x}_0,\mathbf{y}_0\right) - g\left(\mathbf{x}_R,\mathbf{y}_R\right)\right) + \\
& \frac{2}{\mu_g}\left(\frac{2L_{f,\max}}{\lambda} + 2L_{g,\max}\right)\left(v\left(\mathbf{x}_0\right) - v\left(\mathbf{x}_R\right)\right) + \\
& \frac{4Rc\alpha}{\mu_g \lambda^2}\left(\frac{1}{R} + \frac{8\lambda^2 L_{g,\max}^2 c\alpha\zeta_g^2}{\mu_g} + 6\zeta_f^2 + 12\lambda^2\zeta_g^2\right) + \\
& \frac{2Rc\alpha}{\mu_g \lambda^2}\left(8\zeta_f^2 + 8\lambda^2\zeta_g^2\right) + \frac{1}{\lambda^2} + \frac{2R\lambda^2 L_{g,\max}^2 c\alpha\zeta_g^2}{\lambda^2 \mu_g}.
\end{aligned} \tag{52}
$$

Dividing on both sides of the above equation by $R$ results in the following average violation

$$
\begin{aligned}
\frac{\texttt{Viol}_R}{R} \ \leq \ & \frac{2l_f^2}{\mu_g \lambda^2} + \frac{2}{R\mu_g}\left(\frac{2L_{f,\max}}{\lambda^2} + \frac{2L_{g,\max}}{\lambda}\right)\left(f\left(\mathbf{x}_0,\mathbf{y}_0\right) - f\left(\mathbf{x}_R,\mathbf{y}_R\right)\right) + \\
& \frac{2}{R\mu_g}\left(\frac{2L_{f,\max}}{\lambda} + 2L_{g,\max}\right)\left(g\left(\mathbf{x}_0,\mathbf{y}_0\right) - g\left(\mathbf{x}_R,\mathbf{y}_R\right)\right) + \\
& \frac{2}{R\mu_g}\left(\frac{2L_{f,\max}}{\lambda} + 2L_{g,\max}\right)\left(v\left(\mathbf{x}_0\right) - v\left(\mathbf{x}_R\right)\right) + \\
& \frac{4c\alpha}{R\mu_g \lambda^2} + \frac{32L_{g,\max}^2 c^2\alpha^2\zeta_g^2}{\mu_g^2} + \frac{24c\alpha\zeta_f^2}{\mu_g \lambda^2} + \frac{48c\alpha\zeta_g^2}{\mu_g} + \\
& \frac{16c\alpha\zeta_f^2}{\mu_g \lambda^2} + \frac{16c\alpha\zeta_g^2}{\mu_g} + \frac{1}{R\lambda^2} + \frac{2L_{g,\max}^2 c\alpha\zeta_g^2}{\mu_g}.
\end{aligned} \tag{53}
$$

Now, retaining terms that depend on $\lambda$, $R$ and $\alpha$, we get the order result stated in the Theorem. This completes the proof. $\qquad\square$

# H   Examples of $(\alpha, c)-$Robust Aggregators

Though many robust aggregators have been proposed in the literature (Chen et al., 2017; Pillutla et al., 2022b), they do not satisfy the definition 1. Also there are many new attacks where the methods using these aggregators fail to converge.

In this section, we present the aggregators which when used with the bucketing algorithm, introduced in Karimireddy et al. (2021; 2020) satisfy $(\alpha, c)$-robust aggregator definition in 1.

**Geometric Median:** Also known as Robust Federated Averaging (RFA) (Chen et al., 2017; Pillutla et al., 2022b) where aggregation is performed using geometric median:

$$\mathrm{GM}(\mathbf{x}_1, \mathbf{x}_2, \ldots, \mathbf{x}_N) \coloneqq \arg\min_{\mathbf{x} \in \mathbb{R}^d} \sum_{i=1}^{N} \|\mathbf{x} - \mathbf{x}_i\|.$$

**Coordinate-wise Median:** CM is an aggregator which performs co-ordinate wise median:

$$[\mathrm{CM}(\mathbf{x}_1, \mathbf{x}_2, \ldots, \mathbf{x}_N)]_j \coloneqq \mathrm{Median}([\mathbf{x}_1]_j, [\mathbf{x}_2]_j, \ldots, [\mathbf{x}_N]_j),$$

where $[\mathbf{x}]_j$ is the $j^{th}$ component of vector $\mathbf{x}$.

**Krum:** Krum finds a vector $\mathbf{x}_k$ which is closest to the mean of the input vectors when $n - |\mathcal{B}| - 2$ vectors are excluded

$$\mathrm{Krum}(\mathbf{x}_1, \mathbf{x}_2, \ldots, \mathbf{x}_n) \coloneqq \arg\min_{\mathbf{x} \in \{\mathbf{x}_1, \ldots, \mathbf{x}_n\}} \sum_{j \in S_j} \|\mathbf{x}_j - \mathbf{x}_i\|^2.$$

