# OpenReview forum: "Byzantine-Robust and Hessian-Free Federated Bilevel Optimization"
_TMLR — Accepted by TMLR_

### Review · Reviewer_cDCn · 2025-06-25

**Summary Of Contributions:**

This paper addresses the novel and challenging problem of Federated Bilevel Optimization in the presence of Byzantine nodes (FedBOB). The authors note that this problem is largely unexplored, as existing works typically focus on either single-level robust optimization or non-robust federated bilevel optimization.

To overcome the computational complexity of traditional second-order methods and the security risks from malicious nodes, the authors propose Rob-FedBOB, a computationally efficient and robust algorithm. The key innovation is to reformulate the federated bilevel problem into a single-level penalty-based optimization problem, which makes the proposed algorithm fully first-order and Hessian-free. The algorithm is designed to work under more practical and relaxed assumptions, including heterogeneous (non-iid) data across clients and a non-convex lower-level objective that satisfies the Polyak-Łojasiewicz (PL) inequality.

The main contributions of the paper include:

1. Theoretical Performance Analysis: The paper provides a convergence analysis for Rob-FedBOB, showing that the convergence rate depends on the number of communication rounds, the fraction of Byzantine nodes ($\alpha$), and the data heterogeneity. In the absence of Byzantine nodes, the algorithm achieves a convergence rate of $1/R$, where $R$ is the number of communication rounds.

2. New Performance Metric: It introduces and analyzes a new notion of average constraint violation, providing a bound that highlights the trade-off between convergence and constraint satisfaction, managed by the penalty parameter $\lambda$.

3. Experimental Validation: The theoretical findings are empirically corroborated through experiments on applications like data hyper-cleaning and learnable regularization under various Byzantine attacks (e.g., Bit Flipping, Label Flipping).

**Audience:**

Yes

**Broader Impact Concerns:**

N/A.

**Claims And Evidence:**

Yes

**Requested Changes:**

Some minor suggestions:
1. On the top of Page 3, when introducing the results of this paper, the meaning of $\lambda$ is not explained at all, which may confuse readers. The authors are encouraged to add an explanation like "$\lambda$ is a punishment parameter in the algorithm xxx".

2. In the third line of Assumption 2, should it be $v_k$ but not $y_k^*$?

3. On Page 7, it should be $O(\lambda^2/R)$ but not $O(\lambda/R)$.

**Strengths And Weaknesses:**

### Strengths
This problem is interesting. Solving a bilevel optimization problem in the FL setup with non-iid data and PL condition is interesting. This problem is new, although a little artificial.

The paper is well-written and easy to follow.

The authors provided a complete and nice solution to this problem, where the key idea is to leverage inner iterative GD update to approximate the optimal point of the inner optimization problem, at the cost of additional $O(log R)$ communication complexity compared with existing FL methods.

### Weaknesses
The arguments of $O(R\log R)$ communication complexity are a little trick. The authors should also explain when $O(R)$ communication complexity is fixed, this paper can only achieve a near-optimal $O(\log R / R)$-type convergence rate.

As for assumptions, this paper requires $f_k$ and $g_k$ to be Lipschitz and smooth. However, it not only requires $g_k$ to satisfy the PL condition, but also requires the averaged $g$ to satisfy the PL condition. My question is: Is such a requirement on the averaged function also needed in previous works? Some necessary discussions of the assumptions compared with previous works are needed here.

---

### Review · Reviewer_fCue · 2025-07-02

**Summary Of Contributions:**

This work aims to solve federated optimization problem with byzantine nodes. A computationally efficient and robust algorithm using only first order information is proposed. Theoretical guarantees and experiments are provided to validate the proposed method.

**Audience:**

Yes

**Claims And Evidence:**

Yes

**Requested Changes:**

Most issues are related to definition and concept discussion, including

1. The problem of Federated Bilevel Optimization with Byzantine (1) involves upper and lower level objective functions. It would be clearer if the authors can explain what are the upper and lower level objective functions, especially with concrete examples such as in the experiment section. This problem only involves good nodes. I always have the question how Byzantine nodes are involved and impact the optimization?

2. The first paragraph of Problem Statement, 'We consider a federated ... resilient to the Byzantine attacks.' is largely duplicated with the introduction.

3. Can you elaborate how the dual problem in (4) is obtained? Also, more discussion on "solving the above problem is approximately equivalent to solving the original problem" will help readers understand this formulation. Moreover, why (4) is a single level federated learning problem, how do you define a 'single level' optimization problem?

4. About assumptions:  (1) in assumption 2 PL inequality, I was wondering if the first part assumption on each good node implies the second part assumption; (2) the expectation in assumption 3 is taken with respect to what?

5. Can you explain why access to $g(x,y)$ is not available at each node? Is the objective function still well-defined?

6. 'where RAgg uses bucketing followed by geometric median aggregation (see (Karimireddy et al., 2020) for more details).' Same as before, discussion is encouraged for clarity and completeness.

7. At the top of page 6, $k\in[N]$ and $k\in \mathcal{G}$ are not consistent.

8. Can you elaborate why does Lemma 2 shows 'the robustness depends on how close the proxy yr,T is to the actual optimal y∗(xr) captured through the first term in δ2". How does $\delta$ reflect the closeness?

9. The convergence rate is $O(\lambda/R)$ or $O(\lambda^2/R)$? Why the order in (11) does not involve $\lambda$?

10. In lemma 3, the condition $\Vert \nabla h_\lambda(x,y)\Vert \leq \psi$ should hold for any x,y? How strong is this assumption? Moreover, the bounds in Lemma 3 and Theorem 2 do not related to $R$. This means that violation is of constant order and cannot be reduced along the update. Can you provide any insight on this? Why is this result interesting?

**Strengths And Weaknesses:**

The paper is solid. However, I am an outsider of this field and found it hard to understand several points. The writing has room to improve.

---

> ### Author Response · Authors · 2025-07-07
> **Response to Reviewer fCue (Part 1)**
>
> > **Your Comment:** The problem of Federated Bilevel Optimization with Byzantine (1) involves upper and lower level objective functions. It would be clearer if the authors can explain what are the upper and lower level objective functions, especially with concrete examples such as in the experiment section. This problem only involves good nodes. I always have the question how Byzantine nodes are involved and impact the optimization?
>
>  **Our Response:** We thank the reviewer for the comment. Applications such as Data hypercleaning, robust learning are modelled as bilevel optimization problems where they follow a nested structure. Recall, the  data hyper-cleaning application in our experimental section given by
> \begin{equation}
> \min\_{\mathbf x} \frac{1}{Gm}\sum\_{k=1}^G \sum\_{i=1}^m l\_{k,i} \left(\mathbf \ y^* \(\mathbf x); \mathcal{D}\_k^{\text{val}}\right)
> \end{equation}
> subject to
> \begin{eqnarray}
> \mathbf \ y^* \(\mathbf{x}) &\in& \arg \min\_{\mathbf y} \frac{1}{Gn}\sum\_{k=1}^G \sum\_{i=1}^n \sigma(\mathbf x\_{k,i})l\_{k,i}\left(\mathbf y; \mathcal{D}\_k^{\text{train}}\right) + c\left\|\mathbf y \right\|^2.
> \end{eqnarray}
>  Here, $\frac{1}{m} \sum\_{i=1}^m l\_{k,i}\left(\mathbf \ y^* \(\mathbf x); \mathcal{D}\_k^{\text{val}}\right):=f\_k(\mathbf x, \mathbf y^*(x))$ is the upper level objective function and $\frac{1}{n} \sum\_{i=1}^n \sigma(\mathbf x\_{k,i})l\_{k,i}\left(\mathbf y; \mathcal{D}\_k^{\text{train}}\right) + c\left\|\mathbf y \right\|^2 := g\_k(\mathbf x, \mathbf y)$ for all $k \in \mathcal{G}$. The lower level problem aims to find the optimal model parameters $\mathbf y$ that minimizes the weighted average of the loss function (with regularization). Whereas, the upper-level optimization problem aims to find the coefficients $\sigma(\mathbf x\_{k,i})$ by minimizing the validation loss.
>
> **Handling Byzantine Nodes:** Note that the quality of the algorithm is assessed using average loss across "good nodes" as a metric. However, the presence of the Byzantine nodes effect the performance of the algorithm as averaging of the models from all the nodes (good and bad) can happen resulting in a bad model. This is the reason why we use "robust" combining where the effect of byzantine nodes can be mitigated to some extent. It is also important to note that the analysis takes into account the presence of Byzantine nodes ($\alpha$ parameter appears in the result).
>
> > **Your Comment:** The first paragraph of Problem Statement, 'We consider a federated ... resilient to the Byzantine attacks.' is largely duplicated with the introduction.
>
> **Our Response:** We thank the reviewer for the comment. We agree there is some overlap which we will rewrite in the updated manuscript.
>
> > **Your Comment:** Can you elaborate how the dual problem in (4) is obtained? Also, more discussion on "solving the above problem is approximately equivalent to solving the original problem" will help readers understand this formulation. Moreover, why (4) is a single level federated learning problem, how do you define a 'single level' optimization problem?
>
> **Our Response:** We thank the reviewer for the comment. It is well known that the following problem is an approximation of the original bilevel problem stated in the paper (see [1]):
> $$\mathbf P1: \min\_{\mathbf x} \; f\left(\mathbf x, \mathbf y\right)~\nonumber \\
> \textstyle \text{such that} ~p(\mathbf x, \mathbf y) := g\left(\mathbf x, \mathbf y\right) - v(\mathbf x) \leq \epsilon,$$
>
> where $f()$ and $g()$ are the averages of the upper and lower level loss functions from all the "good nodes", respectively. Further, $v(\mathbf x)$ is as defined in the paper. It is important to note that the above problem boils down to the original problem when $\epsilon =0$. The above problem can be recasted as follows (think of Lagrangian formulation):
> $$\texttt{Penalty problem}: \min_{\mathbf x, \mathbf y} h_{\lambda}(\mathbf x, \mathbf y) := f(\mathbf x, \mathbf y) + \lambda (g(\mathbf x, \mathbf y) - v(\mathbf x)),$$
> where $\lambda > 0$ is the penalty factor. By appropriately choosing $\lambda$, the solution to the above problem provides a soluton to the constrianed problem stated above with $\epsilon$ depending on various parameters (see the theorem below). This is made rigorous in Theorem $6$ of [1], which is stated below for convenience.
>
> **Theorem:** Let $(\mathbf x, \mathbf y)$ be a local solution to $\min_{\mathbf x, \mathbf y} \; h_{\lambda}\left(\mathbf x, \mathbf y\right)$. In addition to assumptions $1$-$5$ of the paper, assume that there exists a $\mathbf{\bar{y}}$ such that $p_t(\mathbf x_t, \mathbf{\bar{y}}) \leq \epsilon$, then, $(\mathbf x, \mathbf y)$ is a local optimum solution for the problem $\mathbf P 1$ above with $\epsilon = \frac{L^2}{\lambda^2} \sqrt{\frac{2}{\mu_g}} + 2 \epsilon$.

---

> > ### Author Response · Authors · 2025-07-07
> > **Response to Reviewer fCue (Part 2)**
> >
> > The above theorem states that solving the penalty problem leads to a solution to $\mathbf P1$ with appropriate choice for $\epsilon$. One way to measure the efficiency of the local solution $(\mathbf x, \mathbf y)$ is through $||\nabla h_{\lambda}(\mathbf x, \mathbf y)||^2$.
> >
> > **Single level FL problem:** Note that the $\texttt{Penalty problem}$ corresponds to a single level problem of minimizing the average of functions across nodes.
> >
> > **References:**
> >
> > [1] Han Shen and Tianyi Chen. On penalty-based bilevel gradient descent method. In International Conference on Machine Learning, pages 30992–31015. PMLR, 2023.
> >
> > > **Your Comment:** About assumptions: (1) in assumption 2 PL inequality, I was wondering if the first part assumption on each good node implies the second part assumption; (2) the expectation in assumption 3 is taken with respect to what?
> >
> > **Our Response:** We thank the reviewer for the comment. Recall, the first part of the assumption considers the lower level function $g\_k(\mathbf x, \mathbf y)$ at $k \in \mathbf G$ satisfies PL-inequality whereas second part considers average lower level function $g(\mathbf x, \mathbf y)$ satisfies PL inequality. The first part does not imply the second part. The expectation in $\mathbb{E}\_{k \in \mathcal{G}}\left\|\nabla f\_k \left(\mathbf x, \mathbf y \right)-  \nabla f \left(\mathbf x, \mathbf y \right) \right\|^2 \leq \zeta\_f^2$ is with respect to uniformly sampled nodes in $\mathcal G$, i.e., the set of good nodes. This amounts to $\mathbb{E}\_{k \in \mathcal{G}}\left\|\nabla f_k \left(\mathbf x, \mathbf y \right)-  \nabla f \left(\mathbf x, \mathbf y \right) \right\|^2 = \frac{1}{G} \sum\_{k=1}^G \left\|\nabla f\_k \left(\mathbf x, \mathbf y \right)-  \nabla f \left(\mathbf x, \mathbf y \right) \right\|^2 \leq \zeta\_f^2$. We have used this notation directly from Karimireddy et. al. cited in the Assumption.
> >
> > <!-- We thank the reviewer for the comment. Recall, assumption 1 is about lipschitzness and smoothness of both the upper and lower level functions and assumption 2 is about PL inequality. The smoothness or lipschitz continuity does not imply Pl inequality in general. (2) The expectation is taken over the randomness in the good nodes. -->
> >
> >
> >
> > > **Your Comment:** Can you explain why access to $g(\mathbf x, \mathbf y)$ is not available at each node? Is the objective function still well-defined?
> >
> > **Our Response:** We thank the reviewer for the comment. We have considered a distributed setup where there are multiple clients. Each client has access to only its upper and lower level functions, i.e., $f_k(\mathbf x, \mathbf y)$ and $g_k(\mathbf x, \mathbf y)$ for $k \in [N]$. On the other hand $g(\mathbf x, \mathbf y):= \frac{1}{G} \sum_{k \in \mathcal{G}} g_k(\mathbf x, \mathbf y)$ is the average loss function which is not available at each node. Hence the objective function is well defined. But this function is not accessible to all the nodes, which is typically the case in distributed optimization problems.
> >
> >
> > > **Your Comment:** 'where RAgg uses bucketing followed by geometric median aggregation (see (Karimireddy et al., 2020) for more details).' Same as before, discussion is encouraged for clarity and completeness.
> >
> > **Our Response:** We thank the reviewer for the comment. In the updated mansucript, we have included more explanation to explain why (a) we need robustness and (b) bucketing followed by geometric median aggregation (GMA) brings in more robustness as opposed to GMA.
> >
> > > **Your Comment:** At the top of page 6, $k \in [N]$ and $k \in \mathcal{G}$ are not consistent.
> >
> > **Our Response:** We apologize to the reviewer for the typo. We will correct it in the updated manuscript to include $k \in [N]$.
> >
> > > **Your Comment:** Can you elaborate why does Lemma 2 shows 'the robustness depends on how close the proxy yr,T is to the actual optimal y∗(xr) captured through the first term in δ2". How does $\delta$ reflect the closeness?
> >
> > **Our Response:** The result in Lemma 2 is bounded in terms of $\delta^2$. Recall,
> > $$\delta^2 := \frac{8\lambda^2L^2_{g,\max}l_{g,\max}^2}{\mu_g^2}\left(1-\frac{\gamma \mu_g}{2}\right)^T + \frac{8 \lambda^2 L^2_{g,\max}c \alpha\zeta_g^2}{\mu_g} + 6 \zeta_f^2+ 12 \lambda^2\zeta_g^2.$$
> > The first term in the above update is mostly resulting from bounding $d_{S(\mathbf x_r)}^2(\mathbf y_{r, T})$ which represents how close the proxy $\mathbf y_{r, T}$ is to the optimal $\mathbf y^*(\mathbf x_r)$. The distance is defined as
> > \begin{eqnarray}
> > \textstyle d_{S_t(\mathbf x_t)}(\mathbf y_t) = \min_{\mathbf y_t^{\prime} \in S_t}\left\|\mathbf y_t^{\prime}- \mathbf y_t \right\|,
> > \end{eqnarray}
> > where, $S_t(\mathbf x) := \arg \min_{\mathbf y} g(\mathbf x,\mathbf y)$. Hope this clarifies the doubt.

---

> > > ### Author Response · Authors · 2025-07-07
> > > **Response to Reviewer fCue (Part 3)**
> > >
> > > > **Your Comment:** In lemma 3, the condition $\left\|\nabla  h_\lambda \left(\mathbf x, \mathbf y \right) \right\|^2 \leq \psi$ should hold for any x,y? How strong is this assumption? Moreover, the bounds in Lemma 3 and Theorem 2 do not related to $R$.This means that violation is of constant order and cannot be reduced along the update. Can you provide any insight on this? Why is this result interesting?
> > >
> > > **Our Response:** We thank the reviewer for the comment. First, note that in Theorem 1, we proved a bound on $\left\|\nabla  h_\lambda \left(\mathbf x, \mathbf y \right) \right\|^2$ which is $\Psi := \mathcal{O}\left({\lambda^2}  + \alpha(\zeta_f^2 + \lambda^2 \zeta_g^2)\right)$. Next, we proved a lemma saying if $\left\|\nabla  h_\lambda \left(\mathbf x, \mathbf y \right) \right\|^2 \leq \psi$ holds good for all $\mathbf x, \mathbf y$, then the violation is under control. Therefore, $\left\|\nabla  h_\lambda \left(\mathbf x, \mathbf y \right) \right\|^2 \leq \psi$  is not an assumption.
> > >
> > > Note that $\Psi := \mathcal{O}\left({\lambda^2}  + \alpha(\zeta_f^2 + \lambda^2 \zeta_g^2)\right)$ obtained from Theorem 1 depends on $R$. The result in Theorem 3 depends on $R$ as $1/R$, which is not a dominant term comapred to $\mathcal{O}  \left(\frac{(1+c \alpha \zeta_f^2 )}{\lambda^2} + c \alpha \zeta_g^2\right)$. Hence we have ignored the $1/R$ dependency in the result. The result is interesting in the sense that it characterizes the impact of the byzantine nodes on the average violation constraint. As you rightly pointed out, the average violation is limited by the amount of byzantine nodes in the network. We believe that this cannot be improved further.

---

### Review · Reviewer_eVJZ · 2025-07-02

**Summary Of Contributions:**

The paper proposes an approximate solution to the Federated Bilevel Optimization problem in the presence of Byzantine nodes. The approximate solution utilizes a penalty function to convert the problem into single-level optimization. The authors study the convergence rate in the presence of data heterogeneity and the policy violations. The experimental results study the convergence rate on MNIST for varying number of communication rounds R in the presence of 20% and 40% Byzantine nodes out of 16 total nodes.

**Audience:**

Yes

**Broader Impact Concerns:**

None.

**Claims And Evidence:**

Yes

**Requested Changes:**

In general and as a high priority, I would request the authors to address the weaknesses above.

Additionally, I have some nitpicky concerns which the authors may choose to act on or ignore. This will NOT positively or negatively affect my stand on the paper.

[N1] Split Algorithm 1 into the algorithm for the server and the algorithm for the client. It would make it easier to understand.

[N2] Add attack names to the figures instead of just having them in the captions (Figures 1-4).

[N3] Providing code for the experimental results would be really helpful to the community. Any code is better than no code.

**Strengths And Weaknesses:**

## Strengths

[S1] The paper discusses an important and difficult problem of bilevel optimization in the presence of Byzantine nodes.

[S2] The paper is well written and easy to follow.

[S3] The algorithm is 'second-order free', and hence, computationally efficient.

## Weaknesses

I mainly have concerns with the experimental results section.

[W1] While the authors mention that they ensured the data distribution is heterogeneous, there is no mention of how this was ensured and what level of heterogeneity is exists in the client datasets.

[W2] This is more of a question: I do not understand how the figures 1-4 show the effect of heterogeneity. The text is also not explicit about this.

[W3] The experimental results only show the convergence rates and the policy violations. It would make the paper stronger if the authors also present the actual convergence curves of the task given their method. It would also be nice to how close the results get to the optimal convergence. In this case, comparing against the concurrent Hessian-based federated bilevel optimization solution (Abbas et al., 2024) would also make the results more convincing. Of course, the convergence results do not have to be better than the Hessian-based solution because it is much more computationally expensive. But, having and discussing them would add to the completeness.

[W4] The paper mentions that the approximate problem "boils down to a single level federated learning problem in the presence of Byzantines". I am not convinced why we shouldn't do this. Adding to [W3] above, would it be better or worse experimentally?

---

> ### Author Response · Authors · 2025-07-07
> **Response to Reviewer eVJZ (Part 1)**
>
> > **Your Comment:** While the authors mention that they ensured the data distribution is heterogeneous, there is no mention of how this was ensured and what level of heterogeneity is exists in the client datasets.
>
> **Our Response:** We thank the reviewer for the comment.We have considered total 16 clients out of 3 are byzantine nodes. We assume that byzantine nodes are omniscient and has access to the complete MNIST dataset. On the other hand, the remaining good nodes have combination of samples with 2 labels, for example if client 1 has $0$ and $1$ digits then client 2 has $2$ and $3$ and so on to ensure data heterogeneity across the clients.
>
> > **Your Comment:** This is more of a question: I do not understand how the figures 1-4 show the effect of heterogeneity. The text is also not explicit about this.
>
> **Our Response:** We apologize to the reviewer for the confusion. The figures 1-4 show the convergence rate of the proposed algorithm versus the communication rounds R and the log constraint violation under different attacks for two applications, namely data hypercleaning and learnable regularization. The figures only capture the effect of byzantine nodes ($\alpha$) and does not show the effect of heterogeneity.
>
> > **Your Comment:** The experimental results only show the convergence rates and the policy violations. It would make the paper stronger if the authors also present the actual convergence curves of the task given their method. It would also be nice to how close the results get to the optimal convergence. In this case, comparing against the concurrent Hessian-based federated bilevel optimization solution (Abbas et al., 2024) would also make the results more convincing. Of course, the convergence results do not have to be better than the Hessian-based solution because it is much more computationally expensive. But, having and discussing them would add to the completeness.
>
> **Our Response:** We thank the reviewer for the comment. We will update our response once we get the plots.
>
> > **Your Comment:** The paper mentions that the approximate problem "boils down to a single level federated learning problem in the presence of Byzantines". I am not convinced why we shouldn't do this. Adding to [W3] above, would it be better or worse experimentally?
>
> **Our Response:** We apologize to the reviewer for the confusion. If we understand the comment correctly, the confusion is whether we have used the single level FL problem or not. In fact, we have used the reformulated single level FL problem only. More precisely, we solve the following single level problem in the paper:
> $$\texttt{Penalty problem}: \min_{\mathbf x, \mathbf y} h_{\lambda}(\mathbf x, \mathbf y) := f(\mathbf x, \mathbf y) + \lambda (g(\mathbf x, \mathbf y) - v(\mathbf x)),$$ where $f()$ and $g()$ are the averages of the upper and lower level loss functions from all the "good nodes", respectively. Further, $\lambda > 0$ and $v(\mathbf x)$ is as defined in the paper.

---

> > ### Author Response · Authors · 2025-07-15
> > **Response to Reviewer eVJZ (Part 2)**
> >
> > > **Your Comment:** The experimental results only show the convergence rates and the policy violations. It would make the paper stronger if the authors also present the actual convergence curves of the task given their method. It would also be nice to how close the results get to the optimal convergence. In this case, comparing against the concurrent Hessian-based federated bilevel optimization solution (Abbas et al., 2024) would also make the results more convincing. Of course, the convergence results do not have to be better than the Hessian-based solution because it is much more computationally expensive. But, having and discussing them would add to the completeness.
> >
> > **Our Response:** We thank the reviewer for the comment. The table 1 shows the performance comparison between our proposed method (Rob-FedBOB) and Hessian-based federated bilevel optimization algorithm (BILANTINE) for the data hyper cleaning application. We have considered the bit flipping attack in both the cases. In [1], the authors have demonstrated the superiority of the penalty based method over the second order (Hessian) method in the absence of Byzantine nodes in the centralized case. We on the other hand show superiority of our penalty method compared to BILATINE (second order method) in the FL setting in the presence of Byzantine nodes. As stated in [1], the exact mathematical reason of why the penalty methods work better than the second order methods is not clear, and will be relagated to the future work.
> >
> >
> > | S. No.    |  $R$   | Accuracy (Rob-FedBOB) | Accuracy (BILATINE) |
> > | --- | ---  | ------------- |------------- |
> > |   1  | 1    | 0.66896        | 0.229 |
> > |   2  | 10    |0.71536         | 0.454|
> > |   3  |  50   |    0.756           | 0.66 |
> > |   4  |  75   |  0.76112           | 0.70 |
> > |   5  |  100   |  0.7632           |0.727 |
> > |   6  |  125   |    0.76448           | 0.738 |
> > |   7  |  135   |   0.76448           | 0.753 |
> > |   8  |  150   |   0.7656          | 0.758 |
> >
> > Table 1: Performance comparison of Rob-FedBOB with BILATINE
> >
> > **References:**
> >
> > [1] Kwon, J., Kwon, D., Wright, S. and Nowak, R.D., 2023, July. A fully first-order method for stochastic bilevel optimization. In International Conference on Machine Learning (pp. 18083-18113). PMLR.

---

> > > ### Comment · Reviewer_eVJZ · 2025-07-15
> > >
> > > Thanks to the authors for their response. My concerns have been resolved. I would request the authors to make the required revisions in the submitted paper. Finally, I would like to reiterate that it would be better to provide the code for the experiments to ensure reproducibility.

---

> > > > ### Author Response · Authors · 2025-07-30
> > > >
> > > > Thanks for the positive response. We will be happy to provide the codes and make it public.
> > > >
> > > > Regards,
> > > > Authors

---

### Decision · Action_Editor_SeR2 · 2025-08-20

**Recommendation:** Accept as is

**Audience:**

Yes

**Audience Explanation:**

Bilevel optimization is an important class of problems with a wide range of applications in machine learning, including hyperparameter optimization, meta-learning, neural architecture search, and reinforcement learning. Developing efficient federated algorithms that are robust to Byzantine attacks for bilevel optimization is therefore an important direction that will attract strong interest from the machine learning community and, in particular, the readership of TMLR.

This work presents the first algorithms of this type and establishes convergence guarantees for the proposed methods. Overall, it is a solid study that will be of high interest to the audience.

**Claims And Evidence:**

Yes

**Claims Explanation:**

The theoretical convergence guarantees of the proposed algorithm are rigorously established through mathematical proofs and further validated by experimental results.